# VQEL:
# Enabling Self-Play in Emergent Language Games via Agent-Internal Vector Quantization

## Abstract

Emergent Language (EL) focuses on the emergence of communication among artificial agents. Although symbolic communication channels more closely mirror the discrete nature of human language, learning such protocols remains fundamentally difficult due to the non-differentiability of symbol sampling. Existing approaches typically rely on high-variance gradient estimators such as REINFORCE or on continuous relaxations such as Gumbel–Softmax, both of which suffer from limitations in training stability and scalability. Motivated by cognitive theories that emphasize intrapersonal processes preceding communication, we explore self-play as a substrate for language emergence prior to mutual interaction. We introduce Vector Quantized Emergent Language (VQEL), a novel architecture that incorporates vector quantization into the message generation process. VQEL enables agents to perform self-play using discrete internal representations derived from a learned codebook while preserving end-to-end differentiability. Moreover, the resulting vector-quantized codebook naturally induces a symbolic vocabulary that can be directly transferred and aligned during subsequent mutual play with other agents. Empirical results show that agents pretrained via VQEL self-play achieve more consistent symbol alignment and higher task success when later engaged in mutual interaction. These findings position self-play as a principled and effective mechanism for learning discrete communication protocols, addressing key optimization and representational challenges in emergent language systems.

## 1 Introduction

Emergent Language (EL) studies how communication protocols arise when artificial agents must coordinate to solve cooperative tasks in multi-agent environments Lazaridou et al. (2017); Havrylov & Titov (2017). In such settings, agents typically start without any predefined language and must learn to exchange information to achieve shared objectives. A central motivation of this research is twofold: to shed light on the mechanisms underlying human language evolution and acquisition, and to build artificial systems that can ultimately communicate with humans via natural language Mordatch & Abbeel (2018); Lazaridou et al. (2020); Chaabouni et al. (2021a). For this reason, many EL frameworks explicitly target *symbolic* communication channels. Unlike continuous message vectors, symbolic channels require agents to produce sequences of discrete tokens, better matching the discrete nature of human language Lazaridou et al. (2018a); Bouchacourt & Baroni (2018); Peters et al. (2025).

Training agents to communicate with discrete symbols, however, introduces a core technical obstacle: sampling discrete tokens is non-differentiable. As a result, gradients from the receiver cannot be backpropagated through the sender's discrete decisions. To address this, prior work has largely relied on (i) stochastic gradient estimators such as REINFORCE (Williams, 1992) and (ii) continuous relaxations such as the Gumbel-Softmax estimator (Jang et al., 2017b). However, both approaches suffer from high variance and training instability. Instead of leveraging the informative directional guidance of gradients, they rely on weak scalar rewards, hindering convergence and often leading to suboptimal communication protocols.

Beyond these optimization concerns, we draw loose inspiration from theoretical perspectives in cognitive science and linguistics, which argue that language learning is not only driven by interpersonal exchange, but is also grounded in intrapersonal cognitive processes. Accounts such as the "Language of Thought" hypothesis (Fodor, 1975) and theories of "Inner Speech" (Alderson-Day & Fernyhough, 2015) suggest that agents may first develop internal conceptual structures, a private representational system used to organize experience, before aligning meanings through social interaction. While we do not claim to model these cognitive phenomena directly, they motivate the idea of an agent autonomously forming and stabilizing grounded concepts prior to communication with others.

Motivated by these insights, we investigate whether self-play mechanisms in the context of EL can be leveraged to construct a smoother substrate for language emergence. From this perspective, a self-play–supported EL enables an agent to autonomously form, refine, and stabilize grounded concepts through self-interaction prior to communication with others. Crucially, such a self-play mechanism allows the agent's internal learning to occur directly within the representation space, producing rich learning signals that extend far beyond the sparse, scalar rewards typically employed in methods like REINFORCE. This leads to a central question: how can self-play be effectively instantiated and integrated with mutual play to alleviate the intrinsic difficulty of learning discrete communication protocols?

To answer the above question, we propose a novel architecture based on Vector Quantization (VQ). By integrating VQ into the agent's Message Generation Module, we provide a mechanism that discretizes continuous internal representations into a finite codebook of embedding vectors. This architecture solves the dilemma of Self-Play: it allows the agent to conduct internal games using discrete representations (via the codebook) while maintaining differentiability through the straight-through estimator or commitment losses associated with VQ. Consequently, the agent can "invent" a language internally without the instability of REINFORCE or the continuous relaxation of Gumbel-Softmax. it allows the agent to conduct internal games using discrete representations while maintaining differentiability via VQ commitment losses. By the time the agent engages in mutual play, it has already mapped continuous concepts to discrete symbols. Consequently, while crossing the multi-agent channel still requires policy gradient methods like REINFORCE, the agent does not suffer from the traditional high variance and instability of RL, as it is fine-tuning a structurally sound internal language rather than learning one from scratch. Furthermore, the discrete cluster indices derived from the VQ codebook can be directly mapped to symbols, allowing the internally developed language to be seamlessly transferred and aligned during *Mutual-Play* with other agents.

The contributions of this paper are twofold:

1. We introduce VQEL (Vector Quantized Emergent Language), an architecture that leverages Vector Quantization to facilitate emergent language. This approach provides a stable, gradient-based learning mechanism for discrete communication without relying on REINFORCE or Gumbel-Softmax.

2. We demonstrate the efficacy of Self-Play in emergent language. We show that allowing an agent to develop a foundational language autonomously through VQ-based self-play significantly enhances performance when the agent subsequently engages in Mutual-Play. Through extensive experiments, we illustrate that agents pretrained with self-play achieve better alignment and task success compared to those trained solely through mutual interaction.

## 2 Related work

### 2.1 Emergent Language

Emergent communication studies how artificial agents evolve protocols to solve cooperative tasks without predefined linguistic rules. The standard testbed is the referential game (Lewis signaling game), where a sender communicates a target perception to a receiver Lewis (2008); Lazaridou et al. (2016); Havrylov & Titov (2017). Recent work has expanded this framework to include multi-turn dialogue Evtimova et al. (2018); Jorge et al. (2016); Das et al. (2017); Graesser et al. (2019), population dynamics Ren et al. (2020); Fitzgerald (2019); Chaabouni et al. (2021c), and embodied environments Mordatch & Abbeel (2018). While most research focuses on symbolic transmission, others explore continuous signals Mihai & Hare (2021) or

use communication as a means to solve non-communicative downstream goals rather than as the objective itself Chaabouni et al. (2019); Brandizzi et al. (2022); Eccles et al. (2019).

A primary challenge in symbolic emergent language is the non-differentiability of discrete message channels, which prevents standard backpropagation. Two predominant optimization strategies address this: Policy Gradients and Continuous Relaxations. The REINFORCE algorithm Williams (1992), widely used for its implementation simplicity Foerster et al. (2016); Lazaridou et al. (2016); Bernard & Mickus (2023), treats communication as an action but suffers from high variance and instability Brandizzi (2023). Alternatively, the Gumbel-Softmax relaxation Jang et al. (2017a); Maddison et al. (2016) allows gradients to flow via reparameterization. While Gumbel-Softmax often yields higher performance Havrylov & Titov (2017); Chaabouni et al. (2020); Kharitonov et al. (2020), it relies on continuous approximations during training that deviate from strict discrete communication constraints.

Recent efforts have sought alternatives to these standard paradigms. For example, Carmeli et al. (2023) investigate message quantization, utilizing continuous communication during training and discretizing only during inference. However, this setup creates a discrepancy between training and testing phases and relies on simple scalar quantization. In contrast, our proposed VQEL method enforces discreteness throughout the learning process while maintaining semantic depth, addressing the limitations of prior quantization approaches.

## 2.2 Vector Quantization

Vector Quantization (VQ) is a technique widely used in the domain of signal processing and machine learning. It involves mapping a large set of input vectors into a finite set of output vectors, essentially discretizing continuous input space into a discrete representation space. Vector Quantization discretizes continuous data into discrete representations via a codebook—a finite set of vectors acting as centroids in the embedding space. During encoding, input vectors are mapped to the nearest codebook vector, converting continuous inputs into discrete representations and indices. Optimization refines the embedding space and codebook vectors using techniques like the straight-through estimator and moving average updates of codebook vectors. The most famous and well-known in this field is VQ-VAE, in which VQ is used for image generation Van Den Oord et al. (2017). After the success of VQ-VAE, attention to VQ increased, and it began to be used more in the domains of image and speech. Various modifications were applied to it based on the application and the specific problem. Residual vector quantizationZeghidour et al. (2021), initializing the codebook by the means Zeghidour et al. (2021), having the codebook in a lower dimension Yu et al. (2021), orthogonal regularization loss on codebook Shin et al. (2023), multi-head vector quantizationMama et al. (2021) and expiring stale codes Zeghidour et al. (2021) are some of this works. In this work, we used expiring stale codes to make better use of the codebook and to avoid falling into collapse.

## 3 Method

In traditional research methodologies for investigating emergent language, a significant obstacle is the inability to propagate gradients from the receiver back to the sender during language acquisition and development processes. Although techniques such as REINFORCE and Gumbel–Softmax offer partial solutions, they do not fully model realistic communication environments. Moreover, both approaches rely on approximate gradient estimates for the sender, which can negatively affect optimization and result in reduced accuracy.

Motivated by these limitations, this study aims to investigate the feasibility of enabling an agent to autonomously invent and develop a symbolic communication protocol without requiring interaction with another agent. This is pursued through mechanisms such as internal game-play or self-dialogue. Our proposed solution must meet two essential criteria: First, it should facilitate gradient propagation within the agent itself, distinguishing it from traditional two-agent systems where effective gradient transmission is hindered. Second, the agent must engage in self-interaction using structured linguistic formats, specifically discrete representations, to ensure these representations can be seamlessly employed when interfacing with another agent.

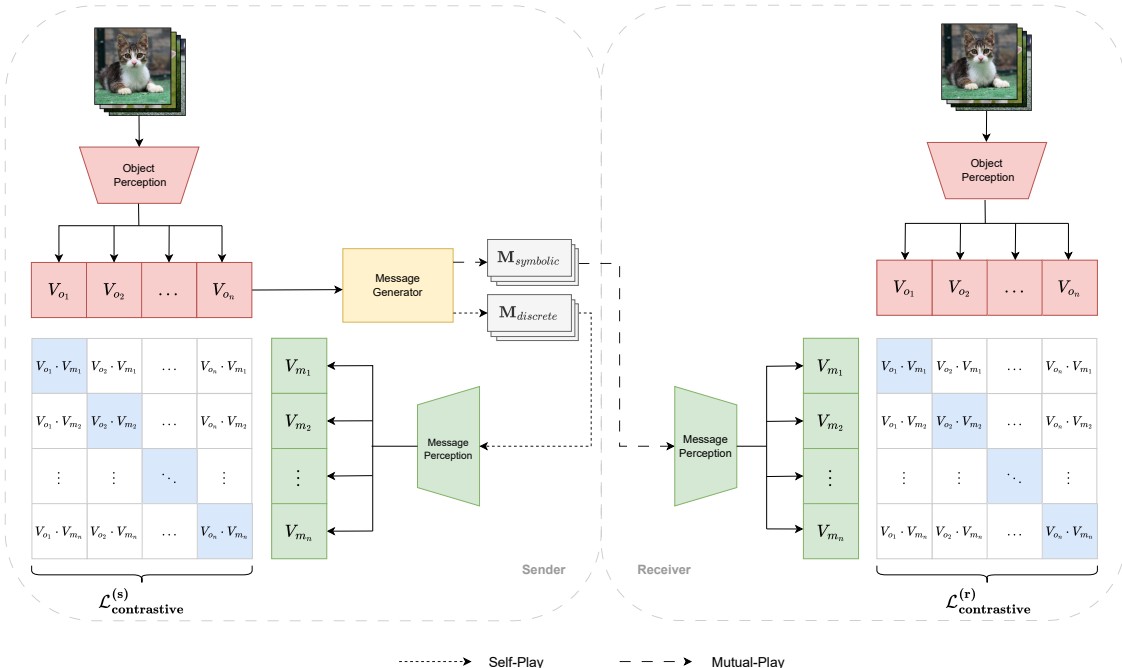

Figure 1: Overview of self-play and mutual play. In self-play, the sender independently develops a symbolic language using a contrastive loss. The sender then interacts with the receiver in mutual play, where the self-developed language is refined through communication using the same contrastive loss.

Drawing inspiration from vector quantization, we focused on embedding this mechanism within the agent, targeting two primary objectives. The first objective is to derive discrete representations conducive to effective gradient-based training during internal dialogue within the agent. The second objective is to utilize these discrete representations as the basis for generating symbols that can be directly applied in interactions with external agents.

## 3.1 Architecture

In our proposed approach, both agents have a similar architecture. Depending on whether they act as a sender or a receiver, they employ different components of this architecture. The architecture of each agent consists of three parts: the **Object Perception Module**, the **Message Generation Module**, and the **Message Perception Module**.

### 3.1.1 Object Perception Module

The Object Perception Module maps an input object $o$ to a continuous vector representation $\mathbf{v}_o \in \mathbb{R}^d$. Formally, this mapping is defined as

$$\mathbf{v}_o = f_{\text{object-perception}}(o), \tag{1}$$

where $f_{\text{object-perception}}$ shows the Object Perception Module that prepares the representation of the object.

### 3.1.2 Message Generation Module

This module takes $\mathbf{v}_o$, the output of the Object Perception Module for object $o$, and returns a sequence of symbols $\mathbf{M}_{\text{symbolic}} = w_1 w_1 \ldots w_L$, where $w_i \in \mathcal{V}$ and $\mathcal{V}$ is the vocabulary of possible symbols (i.e. semantic units or words). Additionally, when used in a self-play scenario, this module outputs a sequence of embedding vectors $\mathbf{M}_{\text{discrete}} = \mathbf{e}_{w_1} \mathbf{e}_{w_1} \ldots \mathbf{e}_{w_L}$ corresponding to the sequence of symbols. This module internally consists of a recurrent neural network and a vector quantization mechanism. This module operates as described in Algorithm 1.

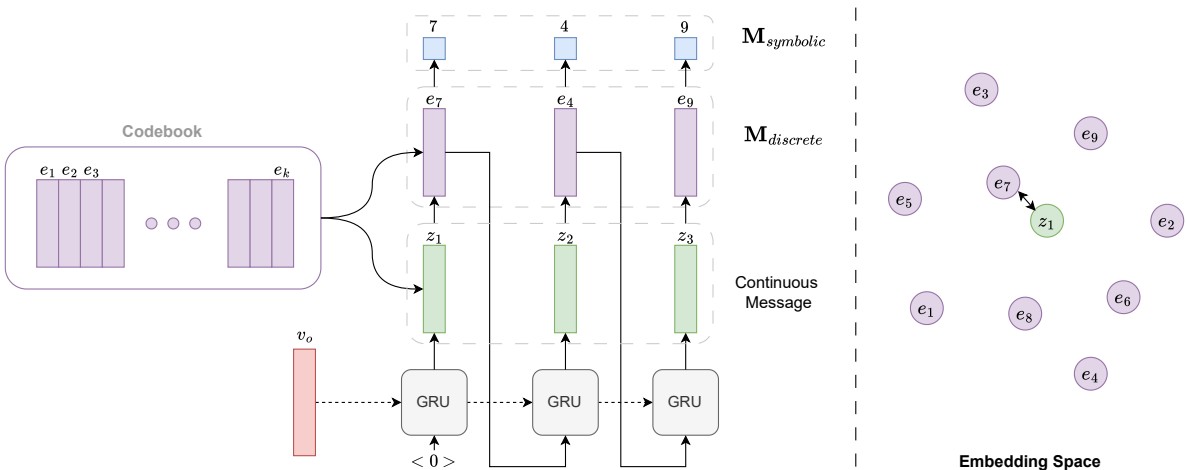

Figure 2: **Left:** Overview of the message generator and the corresponding messages produced during self-play and mutual play. **Right:** Visualization of the embedding space, where the GRU output $\mathbf{z}_1$ is mapped to its nearest codebook embedding $\mathbf{e}_7$.

---

**Algorithm 1** Message Generation Module

---

1: **Input:** $\mathbf{v}_o \in \mathbb{R}^d$; $L$: message length
2: **Module params:** $\mathbf{e}_k \in \mathbb{R}^d, k \in 1, 2, \ldots, K$: codebook; RNN; $g$: linear projection
3:     $\mathbf{h}_0 = \mathbf{v}_o$
4:     **for** $t = 1 \ldots L$
5:         $\mathbf{h}_t = \texttt{RNN}(\mathbf{h}_{t-1}, \texttt{last\_word})$
6:         $\mathbf{z}_t = g(\mathbf{h}_t)$                                    # linear projection of the hidden state layer
7:         $w_t = \arg\min_k \|\mathbf{z}_t - \mathbf{e}_k\|^2$               # hard assignment; see Equation 2 for soft sampling
8:         $\texttt{last\_word} = \mathbf{e}_{w_t}$
9:     $\mathbf{M}_{\text{discrete}} = \{\mathbf{e}_{w_1}, \mathbf{e}_{w_2}, \ldots, \mathbf{e}_{w_L}\}$
10:     $\mathbf{M}_{\text{symbolic}} = \{w_1, w_2, \ldots, w_L\}$
11:     **return** $\mathbf{M}_{\text{discrete}}, \mathbf{M}_{\text{symbolic}}$

---

Alternatively, we can sample $w_t$ from a probability distribution over codebook embeddings, for example using softmax probabilities:

$$P(w_t = k) = \frac{\exp(-\|\mathbf{z}_t - \mathbf{e}_k\|^2 / \tau)}{\sum_{i=1}^{K} \exp(-\|\mathbf{z}_t - \mathbf{e}_i\|^2 / \tau)}, \tag{2}$$

where $\tau$ is a temperature parameter.

### 3.1.3   Message Perception Module

In the mutual-play scenario, this module takes a sequence of symbols $\mathbf{M}_{\text{symbolic}}$ as input and produces a vector representation $\mathbf{v}_m$ for that sequence. When used in a self-play scenario, it takes a sequence of vectors $\mathbf{M}_{\text{discrete}}$ as input and produces a vector representation $\mathbf{v}_m$. Structurally, this module consists of an embedding function and a recurrent neural network:

$$\mathbf{v}_m = \begin{cases} f_{\text{message-perception}}\big(\langle \mathbf{e}_{w_1}, \ldots, \mathbf{e}_{w_L} \rangle\big), & \text{if } \mathbf{M} \text{ is discrete} \\ f_{\text{message-perception}}\big(\langle f_{\text{emb}}(w_1), \ldots, f_{\text{emb}}(w_L) \rangle\big), & \text{if } \mathbf{M} \text{ is symbolic} \end{cases} \tag{3}$$

where $f_{\text{emb}}$ is the embedding function mapping symbols to vectors, and $f_{\text{text-perception}}$ is the recurrent neural network processing the embeddings.

As you have noticed, each of the Message Generation and Message Perception networks mentioned above can have two types of outputs and inputs, respectively: discrete and symbolic. The discrete representations consist of vectors drawn from the learned codebook and are employed during self-play, while the symbolic representations correspond to explicit symbol indices used for communication during mutual play between agents.

### 3.2 Training Algorithm

We have two types of learning processes: **Self-Play** and **Mutual-Play**.

### 3.2.1 Self-Play

In this setting, a single agent plays the referential game with itself, as illustrated for the sender agent in Figure 1. As discussed later, the sender or the receiver can perform the self-play themselves; therefore, we use the superscript $a$ to denote the agent in the following equations. The self-play scenario proceeds as follows.

1. Encode the object to obtain its representation $\mathbf{v}_o^{(a)}$ (Equation 1).

2. Generate the discrete message embeddings $\mathbf{M}_{\text{discrete}}$ from $\mathbf{v}_o^{(a)}$ (Algorithm 1).

3. Encode $\mathbf{M}_{\text{discrete}}$ to obtain the representation of whole message $\mathbf{v}_m^{(a)}$ (Equation 3).

4. Compute the loss function. We use a Contrastive Loss similar to the CLIP loss, defined as:

$$\mathcal{L}_{\text{contrastive}}^{(a)} = -\log \frac{\exp\left(\text{sim}(\mathbf{v}_m^{(a)}, \mathbf{v}_o^{(a)})\right)}{\sum_{o' \in \mathcal{O}} \exp\left(\text{sim}(\mathbf{v}_m^{(a)}, \mathbf{v}_{o'}^{(a)})\right)}, \tag{4}$$

where $\text{sim}(\cdot, \cdot)$ is a similarity function (e.g., the dot product), and $\mathcal{O}$ is the set of objects including the target object and distractors.

We also have a loss related to commitment in vector quantization, which is calculated as follows:

$$\mathcal{L}_{\text{commitment}} = \|\mathbf{z}_t - \text{sg}[\mathbf{e}_{w_t}]\|_2^2, \tag{5}$$

where sg stands for the stopgradient operator. The final self-play loss is given by:

$$\mathcal{L}_{\text{self-play}}^{(a)} = \mathcal{L}_{\text{contrastive}}^{(a)} + \beta \mathcal{L}_{\text{commitment}}, \tag{6}$$

where $\beta$ acts as the weighting factor for the commitment loss, allowing it to be scaled appropriately relative to the contrastive loss.

Since self-play operates over discrete word embeddings (rather than the discrete symbols themselves), we can leverage the straight-through estimator (Van Den Oord et al., 2017) to copy gradients from the discrete representation to its continuous counterpart. This design preserves end-to-end differentiability, allowing all three agent modules to be optimized jointly through backpropagation of $\mathcal{L}_{\text{self-play}}^{(a)}$. The codebook embeddings $\mathcal{C}$ are updated using an exponential moving average procedure following Van Den Oord et al. (2017).

### 3.2.2 Mutual-Play

Two agents play the referential game: one as the sender ($s$) and the other as the receiver ($r$), as shown in Figure 1:

1. **Sender Agent**:

   (a) Encode the object to obtain its vector representation $\mathbf{v}_o^{(s)}$.

   (b) Generate the sequence of symbols $\mathbf{M}_{\text{symbolic}}$, and send it to the receiver agent. During this step, the commitment loss is also calculated according to Equation 5.

2. **Receiver Agent**:

   (a) Encode the received message $\mathbf{M}_{\text{symbolic}}$ to obtain the message embedding $\mathbf{v}_m^{(r)}$.

   (b) Encode the target object and distractors to obtain their vector representations $\mathbf{v}_{o'}^{(r)}$, $\forall o' \in \mathcal{O}$.

   (c) Compute $\mathcal{L}_{\text{contrastive}}^{(r)}$, the contrastive loss for the receiver agent, as defined in Equation 4.

The parameters of the receiver agent's modules (the Message Perception Module and the Object Perception Module) are updated by backpropagating the gradients from $\mathcal{L}_{\text{contrastive}}^{(r)}$. However, the sender cannot receive gradients because the transmission of symbolic messages is inherently non-differentiable. Therefore, we fine-tune the sender using the REINFORCE algorithm:

$$\mathcal{L}_{\text{RL}} = -R \sum_{t=1}^{L} \log P(w_t \mid \mathbf{h}_t), \tag{7}$$

where $P(w_t \mid \mathbf{h}_t)$ is defined in Equation 2. The reward $R$ is based on the receiver's performance and can be formalized as $R = -\mathcal{L}_{\text{contrastive}}^{(r)}$.

As a result, the overall loss for mutual play is:

$$\mathcal{L}_{\text{mutual-play}} = \mathcal{L}_{\text{contrastive}}^{(r)} + \mathcal{L}_{\text{RL}} + \beta \, \mathcal{L}_{\text{commitment}}, \tag{8}$$

where the receiver is updated by the first term, and the sender is updated by the second and third terms.

In the overall process, performing the self-play game before the mutual-play game allows the agent to leverage end-to-end gradient learning to build a better internal and foundational language for communication with other agents.

## 4 Experimental Setup

### 4.1 Datasets

**Synthetic Objects.** This dataset is based on EGG's object game (Kharitonov et al., 2019), and designed to cover the full space of categorical attribute combinations. It contains 10,000 unique objects, each defined by four categorical attributes with ten possible values each. Objects are represented as 40-dimensional vectors formed by concatenating four one-hot encodings. A key challenge of this dataset is that the inputs are discrete, in contrast to datasets with continuous or visual representations.

**ShapeWorld.** This dataset consists of synthetic images of single geometric objects rendered in different colors on a black background (Kuhnle & Copestake, 2017). It enables explicit control over compositional structure. In our setup, the training and test splits differ in compositionality: some color–shape combinations are seen during training, while others are held out and appear only at test time. Consequently, all reported ShapeWorld accuracies measure compositional, out-of-distribution generalization to unseen color–shape combinations.

**DSprites.** It is a synthetic dataset for studying disentangled representations (Matthey et al., 2017). It includes 737,280 black-and-white $64 \times 64$ images generated by varying five latent factors: shape, scale, rotation, and x- and y-position. Its explicitly structured latent space makes it well suited for evaluating disentanglement through emergent communication.

**CelebA.** This dataset contains 202,599 face images of size $178 \times 218$ pixels from 10,177 identities, each annotated with 40 binary facial attributes (Liu et al., 2015). Compared to the synthetic datasets, CelebA introduces the additional complexity of natural image data.

### 4.2 Variations of the Proposed Model

**Sender Self-Play.** In this version, the sender first undergoes self-play before interacting with the receiver in the mutual-play. During the mutual-play, the sender can either be frozen, fine-tuned using REINFORCE (RL) (see Equation 7), or fine-tuned using both the REINFORCE and self-play objectives (see Equation 6) (RL+Pres). In the first scenario, the sender's language remains fixed; in the second, it is optimized for communication; and in the third, it improves for communication while attempting to preserve its original language.

**Sender and Receiver Self-Play.** We designed an experiment in which each agent first invents its own language during the self-play and then communicates with the other agent in the mutual-play to converge to a shared language. To encourage the development of different languages during self-play, agents are initialized with different seeds.

**Receiver Self-Play.** We also conducted a receiver self-play experiment, in which the receiver first undergoes self-play before interacting with the sender in mutual-play. Since the codebook used for communication is constructed by the sender, this setting acts as a natural control: self-play on the receiver — which does not own the codebook — is not expected to transfer, and indeed yields no benefit. We report and analyze these results in detail in Appendix B.

### 4.3 Baselines

We compare VQEL against two commonly used baselines for emergent communication in referential games: REINFORCE (Williams, 1992) and GS (Jang et al., 2017b; Maddison et al., 2016). To maintain backwards differentiability in GS, we use the straight-through (ST) trick (Havrylov & Titov, 2017) with learning the inverse-temperature with a multilayer perceptron (Havrylov & Titov, 2017):

$$\frac{1}{\tau(h_i)} = \log(1 + \exp(\mathbf{w}_\tau^T \mathbf{h}_i)) + \tau_0 \tag{9}$$

where $\tau_0$ controls maximum possible value for the temperature.

### 4.4 Agents' Architecture

**The Object Perception Module** is a simple embedding layer for the Objects dataset, and a simple CNN encoder architecture adopted from Prototypical Networks (Snell et al., 2017) for ShapeWorld and dSprites. In contrast, for CelebA we use a small pretrained DINOv2 network (Oquab et al., 2023) with a linear layer on top. The DINOv2 network's parameters are frozen during training, and only the linear layer is trained.

**The Message Generation Module** comprises a GRU and a VQ module. In the VQ module, we use cosine similarity as the distance metric. We observe that constraining the code vectors to lie on a hypersphere leads to improved communication success. For completeness, we also report results obtained using Euclidean distance in Appendix A.

**The Message Perception Module** consists of a GRU and an embedding layer, which converts symbolic messages into embeddings before passing them through the GRU.

All three methods share the same overall architecture, except that GS-ST and REINFORCE do not include VQ module. The number of parameters is identical across all methods.

### 4.5 Evaluation Metrics

We evaluate all experiments using four metrics. **Accuracy (ACC)** measures communication success as the fraction of correctly identified target objects. **Active Words (AW)** (Lazaridou et al., 2016) represents the fraction of the vocabulary that is used at least once during communication. **Topographic Similarity (TopSim)** (Brighton & Kirby, 2006; Lazaridou et al., 2018b) measures the structural alignment between attribute representations and generated messages, computed as the correlation between Hamming distances in the attribute space and message space. **Entropy of the concept given the message, $H(C \mid M)$** (Rita et al., 2021), measures the uncertainty over concepts conditioned on the received message.

### 4.6 Training Details

In each experiment, the dataset is split into training, validation, and test sets with proportions of 80%, 10%, and 10%, respectively. All methods use the Adam optimizer with a weight decay of $1 \times 10^{-5}$. The learning rate is tuned by searching the range $[10^{-6}, 10^{-3}]$ with a step size of 0.1. The sampling temperature is optimized over the range $[10^{-5}, 1]$ with a step size of 0.1, and $\tau_0$ is tuned by selecting the best value from $[0.1, 1.5]$ with a step size of 0.1.

Baseline methods are trained for 100 epochs, while VQEL is trained for 50 epochs in the self-play phase, followed by an additional 50 epochs in mutual play. The vocabulary size is set to 10, and the message length $L$ is 4 in all experiments. During training, agents are trained with a batch size of 32 (corresponding to 31 distractors), whereas at test time, the main results are reported using a batch size of 100. Additionally, we evaluate methods under different batch sizes to analyze their robustness.

## 5 Results

For simplicity, in the following tables we denote self-play and mutual play as SP and MP, respectively. The agent performing self-play is indicated in subscript; for example, $SP_S$ refers to sender self-play. We use the + notation to indicate sequential training phases (e.g., $SP_S + MP$ denotes sender self-play followed by mutual play). Entries denoted solely as SP (e.g., VQEL-$SP_S$) report the performance of the agent's internal language established during the self-play phase, prior to any mutual interaction.

To preserve comparability, the GS-ST and REINFORCE results are copied from Table 1 into the subsequent tables. In these tables, the highest accuracy for each setting is highlighted in bold, and when relevant, the second-highest accuracy is also indicated to facilitate comparison between the most competitive results.

### 5.1 Sender Self-Play

The results are shown in Table 1. Across all datasets except CelebA, the accuracy of all three modes exceeds that of both REINFORCE and GS-ST. Notably, the ShapeWorld results are measured on a held-out compositional split, where some color–shape combinations appear only at test time; VQEL's improvement on this dataset therefore reflects stronger compositional, out-of-distribution generalization to unseen combinations rather than mere in-distribution fitting. Moreover, allowing the sender to improve its language during communication consistently increases accuracy. For CelebA, REINFORCE achieves the best performance, likely due to the use of pretrained encoders and the limited portion of the model that can take advantage of our method. However, as shown in Section 5.2, there exists another scenario in which our method even outperforms REINFORCE on CelebA. As shown in Figure 3, VQEL demonstrates greater robustness to an increasing number of distractors during testing, with a significantly smaller decline in accuracy relative to the baseline methods.

Interestingly, although the vocabulary size is limited to 10, both GS-ST and REINFORCE fail to utilize the full vocabulary (AW < 1). In contrast, VQEL fully exploits 100% of the vocabulary across all datasets and modes. Additionally, our method yields substantially lower entropy, demonstrating more efficient message encoding.

As previous works (Yao et al., 2022; Chaabouni et al., 2021b) have shown no clear relationship between accuracy and TopSim metric, we observe that VQEL can sometimes improve TopSim, while in other cases it remains comparable to REINFORCE. A more detailed analysis of the learned communication structures and token-level compositionality is provided in Appendix C.

Finally, as illustrated in Figure 4, which depicts the number of unique messages generated by different methods across datasets, Our method more effectively utilizes channel capacity. Consequently, it generates a higher number of unique messages compared to both REINFORCE and GS-ST.

| Dataset | Method | Sender Update | ACC $\uparrow$ | AW $\uparrow$ | TopSim $\uparrow$ | H(C\|M) $\downarrow$ |
|---|---|---|---|---|---|---|
| OBJECTS | GS-ST | - | $0.78_{\pm0.01}$ | $0.93_{\pm0.06}$ | $0.21_{\pm0.02}$ | $1.04_{\pm0.03}$ |
| | REINFORCE | - | $0.51_{\pm0.21}$ | $0.47_{\pm0.12}$ | $0.14_{\pm0.07}$ | $2.21_{\pm1.28}$ |
| | VQEL-SP$_S$ | - | $0.82_{\pm0.01}$ | $1.00_{\pm0.00}$ | $0.19_{\pm0.01}$ | $0.12_{\pm0.02}$ |
| | VQEL-SP$_S$+MP | Frozen | $\underline{0.85_{\pm0.01}}$ | $1.00_{\pm0.00}$ | $0.19_{\pm0.01}$ | $0.12_{\pm0.02}$ |
| | VQEL-SP$_S$+MP | RL | $\mathbf{0.86_{\pm0.01}}$ | $1.00_{\pm0.00}$ | $0.19_{\pm0.01}$ | $0.12_{\pm0.02}$ |
| | VQEL-SP$_S$+MP | RL+Pres | $\mathbf{0.86_{\pm0.01}}$ | $1.00_{\pm0.00}$ | $0.19_{\pm0.01}$ | $0.12_{\pm0.02}$ |
| SHAPE | GS-ST | - | $0.82_{\pm0.01}$ | $1.00_{\pm0.00}$ | $0.02_{\pm0.01}$ | $1.21_{\pm0.11}$ |
| | REINFORCE | - | $0.86_{\pm0.00}$ | $0.83_{\pm0.15}$ | $0.01_{\pm0.01}$ | $0.88_{\pm0.11}$ |
| | VQEL-SP$_S$ | - | $0.87_{\pm0.01}$ | $1.00_{\pm0.00}$ | $0.05_{\pm0.03}$ | $0.38_{\pm0.10}$ |
| | VQEL-SP$_S$+MP | Frozen | $0.89_{\pm0.01}$ | $1.00_{\pm0.00}$ | $0.05_{\pm0.03}$ | $0.38_{\pm0.10}$ |
| | VQEL-SP$_S$+MP | RL | $\mathbf{0.91_{\pm0.01}}$ | $1.00_{\pm0.00}$ | $0.05_{\pm0.02}$ | $0.39_{\pm0.10}$ |
| | VQEL-SP$_S$+MP | RL+Pres | $\underline{0.91_{\pm0.02}}$ | $1.00_{\pm0.00}$ | $0.05_{\pm0.03}$ | $0.39_{\pm0.10}$ |
| DSPRITES | GS-ST | - | $0.81_{\pm0.01}$ | $0.90_{\pm0.00}$ | $0.10_{\pm0.01}$ | $1.80_{\pm0.05}$ |
| | REINFORCE | - | $0.88_{\pm0.02}$ | $0.80_{\pm0.10}$ | $0.06_{\pm0.00}$ | $1.06_{\pm0.13}$ |
| | VQEL-SP$_S$ | - | $0.91_{\pm0.02}$ | $1.00_{\pm0.00}$ | $0.07_{\pm0.01}$ | $0.40_{\pm0.02}$ |
| | VQEL-SP$_S$+MP | Frozen | $\underline{0.92_{\pm0.01}}$ | $1.00_{\pm0.00}$ | $0.07_{\pm0.01}$ | $0.40_{\pm0.02}$ |
| | VQEL-SP$_S$+MP | RL | $\mathbf{0.93_{\pm0.01}}$ | $1.00_{\pm0.00}$ | $0.07_{\pm0.01}$ | $0.40_{\pm0.01}$ |
| | VQEL-SP$_S$+MP | RL+Pres | $\mathbf{0.93_{\pm0.01}}$ | $1.00_{\pm0.00}$ | $0.07_{\pm0.01}$ | $0.40_{\pm0.02}$ |
| CELEBA | GS-ST | - | $0.90_{\pm0.00}$ | $1.00_{\pm0.00}$ | $0.14_{\pm0.01}$ | $1.01_{\pm0.08}$ |
| | REINFORCE | - | $\mathbf{0.93_{\pm0.01}}$ | $1.00_{\pm0.00}$ | $0.11_{\pm0.03}$ | $0.90_{\pm0.06}$ |
| | VQEL-SP$_S$ | - | $0.89_{\pm0.01}$ | $1.00_{\pm0.00}$ | $0.10_{\pm0.04}$ | $0.58_{\pm0.10}$ |
| | VQEL-SP$_S$+MP | Frozen | $0.90_{\pm0.01}$ | $1.00_{\pm0.00}$ | $0.10_{\pm0.04}$ | $0.58_{\pm0.10}$ |
| | VQEL-SP$_S$+MP | RL | $\underline{0.91_{\pm0.01}}$ | $1.00_{\pm0.00}$ | $0.10_{\pm0.04}$ | $0.57_{\pm0.09}$ |
| | VQEL-SP$_S$+MP | RL+Pres | $\underline{0.91_{\pm0.01}}$ | $1.00_{\pm0.00}$ | $0.10_{\pm0.04}$ | $0.56_{\pm0.09}$ |

Table 1: Performance comparison across datasets and evaluation metrics for the sender self-play game.

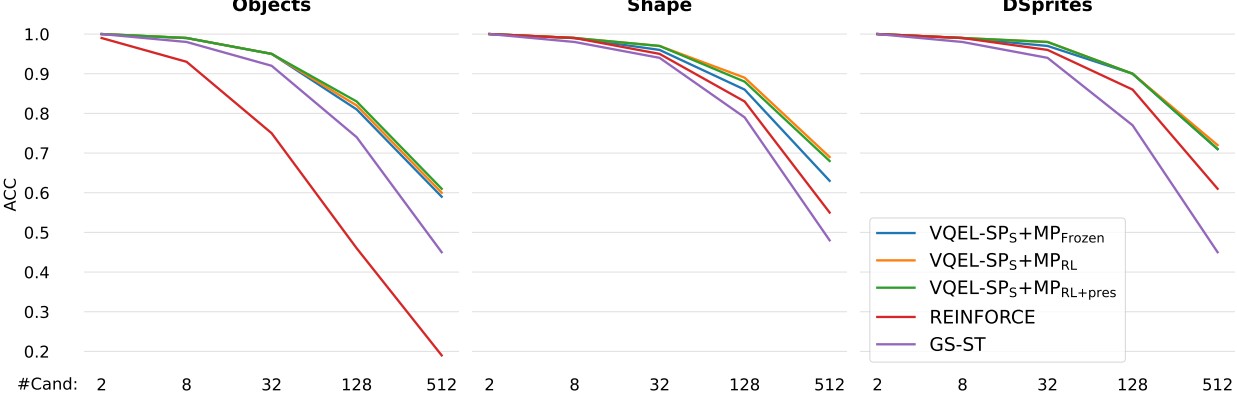

Figure 3: Accuracy of the sender self-play game compared to baseline methods for varying numbers of candidates at test time.

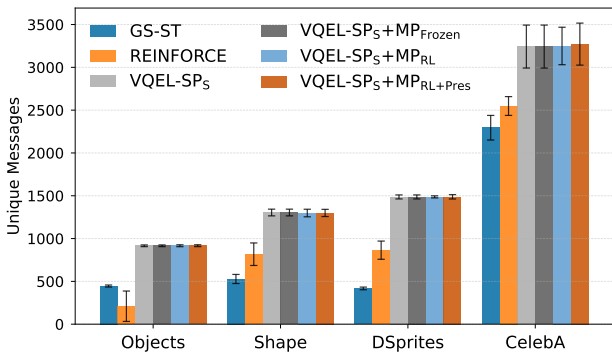

Figure 4: Comparison of the number of unique messages produced by VQEL, during the sender self-play game, and baseline models.

## 5.2 Sender and Receiver Self-Play

The results for this game, shown in Table 2, demonstrate that our method outperforms both REINFORCE and GS-ST across all four datasets. Notably, for the Objects and CelebA datasets, it also surpasses VQEL in the Sender Self-Play scenario (Section 5.1).

Furthermore, Figure 5 illustrates that VQEL maintains higher accuracy than the baselines as the number of test-time candidates increases, exhibiting a smaller performance drop.

## 5.3 Effect of Self-Play

The improvements in communication observed in the previous experiments raise an important question: are these gains due to the self-play technique, or are they primarily the result of using vector quantization in the message generator? To investigate this, we designed an experiment in which the model is trained for the full 100 epochs in the mutual-play scenario without any self-play. As shown in Table 3, removing self-play leads to a significant drop in accuracy, highlighting the crucial role of inventing a symbolic language during the self-play phase.

The instability of learning from scratch is most evident on the Objects dataset, where both REINFORCE $(0.51_{\pm0.21})$ and VQEL-MP $(0.28_{\pm0.18})$ in Table 3 exhibit low accuracy together with high variance. Unlike the visual datasets, the inputs of Objects are discrete one-hot vectors with no gradual similarity structure between objects, so the agent must learn an essentially arbitrary mapping from a large discrete input space onto a discrete protocol — precisely the regime in which the high-variance, weak scalar reward of REINFORCE struggles most. Depending on the random seed, optimization sometimes discovers a workable protocol and sometimes collapses into a degenerate one, which explains the large standard deviation. VQEL-MP is even

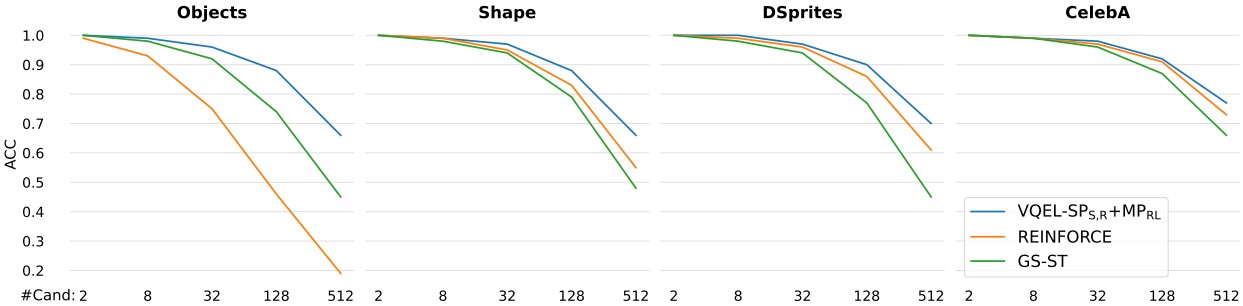

Figure 5: Accuracy of the sender and receiver self-play game compared to baseline methods for varying numbers of candidates at test time.

| Dataset | Method | ACC ↑ | AW ↑ | TopSim ↑ | H(C\|M) ↓ |
|---------|--------|-------|------|----------|-----------|
| OBJECTS | GS-ST | $0.78_{\pm 0.01}$ | $0.93_{\pm 0.06}$ | $0.21_{\pm 0.02}$ | $1.04_{\pm 0.03}$ |
| | REINFORCE | $0.51_{\pm 0.21}$ | $0.47_{\pm 0.12}$ | $0.14_{\pm 0.07}$ | $2.21_{\pm 1.28}$ |
| | VQEL-SP$_S$ | $0.82_{\pm 0.01}$ | $1.00_{\pm 0.00}$ | $0.19_{\pm 0.01}$ | $0.12_{\pm 0.02}$ |
| | VQEL-SP$_R$ | $0.81_{\pm 0.01}$ | $1.00_{\pm 0.00}$ | $0.19_{\pm 0.01}$ | $0.12_{\pm 0.00}$ |
| | VQEL-SP$_{S,R}$+MP | $\mathbf{0.90_{\pm 0.01}}$ | $1.00_{\pm 0.00}$ | $0.17_{\pm 0.02}$ | $0.14_{\pm 0.02}$ |
| SHAPE | GS-ST | $0.82_{\pm 0.01}$ | $1.00_{\pm 0.00}$ | $0.02_{\pm 0.01}$ | $1.21_{\pm 0.11}$ |
| | REINFORCE | $0.86_{\pm 0.00}$ | $0.83_{\pm 0.15}$ | $0.01_{\pm 0.01}$ | $0.88_{\pm 0.11}$ |
| | VQEL-SP$_S$ | $0.87_{\pm 0.01}$ | $1.00_{\pm 0.00}$ | $0.05_{\pm 0.03}$ | $0.38_{\pm 0.10}$ |
| | VQEL-SP$_R$ | $0.87_{\pm 0.01}$ | $1.00_{\pm 0.00}$ | $0.05_{\pm 0.03}$ | $0.38_{\pm 0.10}$ |
| | VQEL-SP$_{S,R}$+MP | $\mathbf{0.91_{\pm 0.00}}$ | $1.00_{\pm 0.00}$ | $0.04_{\pm 0.01}$ | $0.42_{\pm 0.02}$ |
| DSPRITES | GS-ST | $0.81_{\pm 0.01}$ | $0.90_{\pm 0.00}$ | $0.10_{\pm 0.01}$ | $1.80_{\pm 0.05}$ |
| | REINFORCE | $0.88_{\pm 0.02}$ | $0.80_{\pm 0.10}$ | $0.06_{\pm 0.00}$ | $1.06_{\pm 0.13}$ |
| | VQEL-SP$_S$ | $0.91_{\pm 0.02}$ | $1.00_{\pm 0.00}$ | $0.07_{\pm 0.01}$ | $0.40_{\pm 0.02}$ |
| | VQEL-SP$_R$ | $0.90_{\pm 0.01}$ | $1.00_{\pm 0.00}$ | $0.07_{\pm 0.01}$ | $0.38_{\pm 0.03}$ |
| | VQEL-SP$_{S,R}$+MP | $\mathbf{0.92_{\pm 0.01}}$ | $1.00_{\pm 0.00}$ | $0.09_{\pm 0.00}$ | $0.45_{\pm 0.04}$ |
| CELEBA | GS-ST | $0.90_{\pm 0.00}$ | $1.00_{\pm 0.00}$ | $0.14_{\pm 0.01}$ | $1.01_{\pm 0.08}$ |
| | REINFORCE | $0.93_{\pm 0.01}$ | $1.00_{\pm 0.00}$ | $0.11_{\pm 0.03}$ | $0.90_{\pm 0.06}$ |
| | VQEL-SP$_S$ | $0.89_{\pm 0.01}$ | $1.00_{\pm 0.00}$ | $0.10_{\pm 0.04}$ | $0.58_{\pm 0.10}$ |
| | VQEL-SP$_R$ | $0.89_{\pm 0.00}$ | $1.00_{\pm 0.00}$ | $0.11_{\pm 0.04}$ | $0.58_{\pm 0.12}$ |
| | VQEL-SP$_{S,R}$+MP | $\mathbf{0.94_{\pm 0.00}}$ | $1.00_{\pm 0.00}$ | $0.11_{\pm 0.01}$ | $0.53_{\pm 0.07}$ |

Table 2: Performance comparison across datasets and evaluation metrics for the sender and receiver self-play game.

| Dataset | Method | Sender Update | ACC ↑ | AW ↑ | TopSim ↑ | H(C\|M) ↓ |
|---------|--------|---------------|-------|------|----------|-----------|
| OBJECTS | GS-ST | - | $0.78_{\pm 0.01}$ | $0.93_{\pm 0.06}$ | $0.21_{\pm 0.02}$ | $1.04_{\pm 0.03}$ |
| | REINFORCE | - | $0.51_{\pm 0.21}$ | $0.47_{\pm 0.12}$ | $0.14_{\pm 0.07}$ | $2.21_{\pm 1.28}$ |
| | VQEL-SP$_S$+MP | RL | $\mathbf{0.86_{\pm 0.01}}$ | $1.00_{\pm 0.00}$ | $0.19_{\pm 0.01}$ | $0.12_{\pm 0.02}$ |
| | VQEL-MP | RL | $0.28_{\pm 0.18}$ | $1.00_{\pm 0.00}$ | $0.22_{\pm 0.04}$ | $2.64_{\pm 0.32}$ |
| SHAPE | GS-ST | - | $0.82_{\pm 0.01}$ | $1.00_{\pm 0.00}$ | $0.02_{\pm 0.01}$ | $1.21_{\pm 0.11}$ |
| | REINFORCE | - | $0.86_{\pm 0.00}$ | $0.83_{\pm 0.15}$ | $0.01_{\pm 0.01}$ | $0.88_{\pm 0.11}$ |
| | VQEL-SP$_S$+MP | RL | $\mathbf{0.91_{\pm 0.01}}$ | $1.00_{\pm 0.00}$ | $0.05_{\pm 0.02}$ | $0.39_{\pm 0.10}$ |
| | VQEL-MP | RL | $0.88_{\pm 0.01}$ | $1.00_{\pm 0.00}$ | $0.01_{\pm 0.00}$ | $0.67_{\pm 0.18}$ |
| DSPRITES | GS-ST | - | $0.81_{\pm 0.01}$ | $0.90_{\pm 0.00}$ | $0.10_{\pm 0.01}$ | $1.80_{\pm 0.05}$ |
| | REINFORCE | - | $0.88_{\pm 0.02}$ | $0.80_{\pm 0.10}$ | $0.06_{\pm 0.00}$ | $1.06_{\pm 0.13}$ |
| | VQEL-SP$_S$+MP | RL | $\mathbf{0.93_{\pm 0.01}}$ | $1.00_{\pm 0.00}$ | $0.07_{\pm 0.01}$ | $0.40_{\pm 0.01}$ |
| | VQEL-MP | RL | $0.88_{\pm 0.02}$ | $1.00_{\pm 0.00}$ | $0.05_{\pm 0.01}$ | $0.63_{\pm 0.09}$ |
| CELEBA | GS-ST | - | $0.90_{\pm 0.00}$ | $1.00_{\pm 0.00}$ | $0.14_{\pm 0.01}$ | $1.01_{\pm 0.08}$ |
| | REINFORCE | - | $\mathbf{0.93_{\pm 0.01}}$ | $1.00_{\pm 0.00}$ | $0.11_{\pm 0.03}$ | $0.90_{\pm 0.06}$ |
| | VQEL-SP$_S$+MP | RL | $\underline{0.91_{\pm 0.01}}$ | $1.00_{\pm 0.00}$ | $0.10_{\pm 0.04}$ | $0.57_{\pm 0.09}$ |
| | VQEL-MP | RL | $0.87_{\pm 0.03}$ | $1.00_{\pm 0.00}$ | $0.13_{\pm 0.02}$ | $0.74_{\pm 0.10}$ |

Table 3: Effect of self-play on the agents' communication performance.

more fragile than REINFORCE here, because without a self-play phase to first stabilize the codebook it must learn both a meaningful codebook and the object-to-symbol mapping simultaneously, using only the same high-variance signal. By contrast, the same architecture preceded by self-play (VQEL-SP$_S$+MP) attains $0.86_{\pm 0.01}$ on Objects in Table 3 — both higher and far more stable — directly confirming that the self-play phase, rather than the VQ module alone, is what resolves this instability.

To further illustrate this stabilizing effect across the training process, Figure 6 plots the training accuracy over multiple random seeds. As shown, VQEL demonstrates a significantly smoother learning trajectory

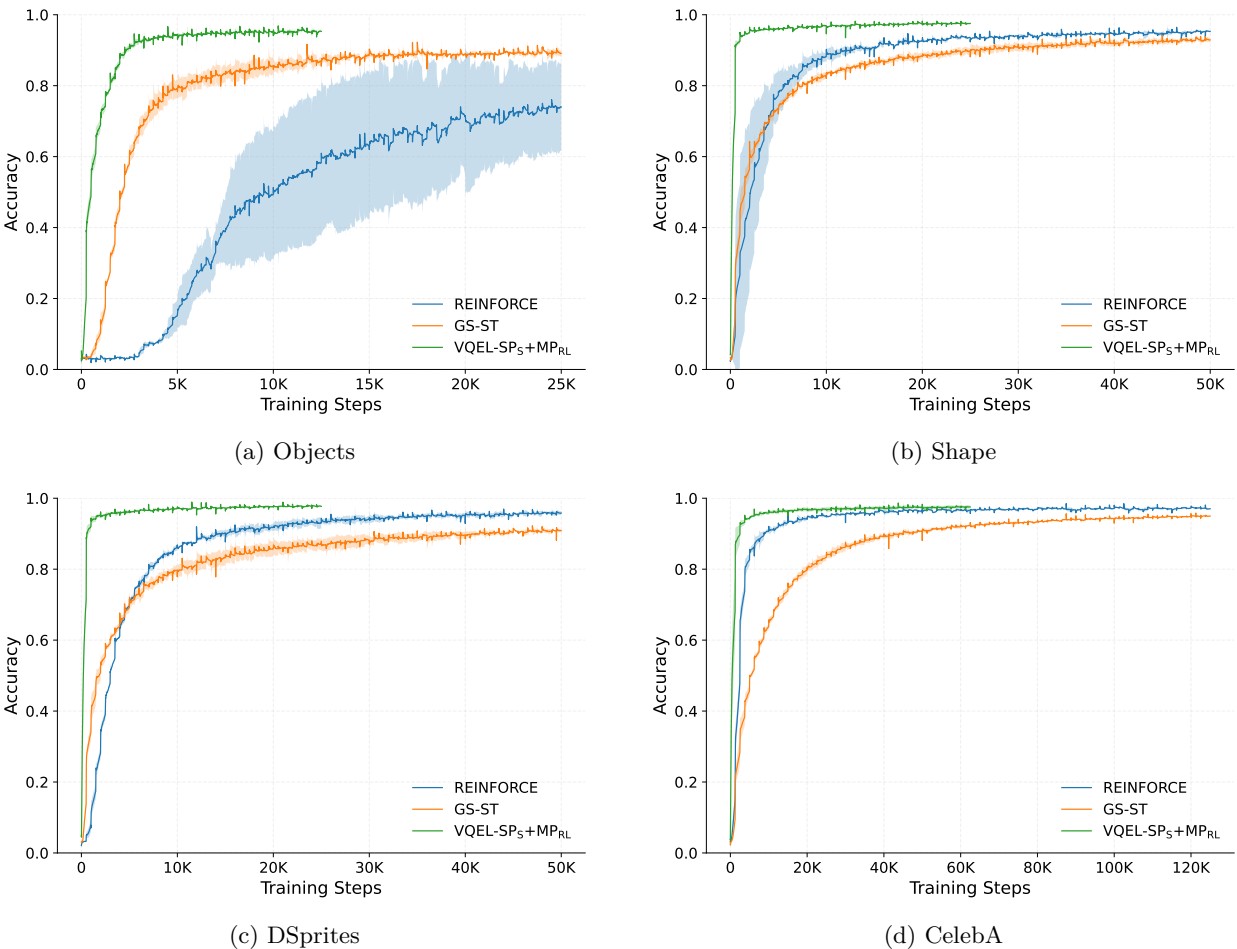

(a) Objects

(b) Shape

(c) DSprites

(d) CelebA

Figure 6: Training stability across four datasets. Curves show the mean accuracy over multiple random seeds, and shaded regions denote one standard deviation. Our method demonstrates reduced variance on Objects and Shape, while exhibiting comparable stability to the baselines on DSprites and CelebA.

and reduced variance compared to the baselines, particularly on the Objects and Shape datasets. While the stability of all methods is more comparable on dSprites and CelebA, VQEL avoids the severe performance drops and high variance seen in REINFORCE, empirically confirming that the self-play phase provides a much more robust optimization landscape for the subsequent mutual-play.

## 5.4 Effect of Vector Quantization

The ablation in the previous section showed that removing the self-play stage substantially degrades performance, demonstrating that self-play is a key component of VQEL. A natural question, however, is whether the observed improvements stem simply from pretraining or whether the VQ module, which provides two complementary representations of each token—a discrete representation for self-play and a symbolic representation for mutual play—is itself essential. To answer this question, we compare VQEL against a GS-ST baseline that is pretrained using the same self-play procedure.

Specifically, we apply the identical self-play protocol to the GS-ST sender. During self-play, the sender communicates only with itself, eliminating the requirement for a symbolic communication channel. We therefore evaluate both Gumbel-Softmax variants during self-play: the standard differentiable formulation

and the Straight-Through (ST) estimator. After self-play, the pretrained sender is jointly optimized with the receiver during the standard mutual-play stage.

Table 4 summarizes the results. Neither self-play pretraining strategy improves upon the GS-ST baseline, whereas VQEL consistently achieves superior performance. These findings indicate that the benefits of VQEL cannot be attributed to self-play as a generic pretraining procedure. Instead, they suggest that the VQ module is a crucial component, as it enables two complementary representations of each token: a discrete representation that facilitates effective self-play and a symbolic representation that supports communication during mutual play.

| Dataset | Method | SP Training | ACC ↑ | AW ↑ | TopSim ↑ | H(C\|M) ↓ |
|---|---|---|---|---|---|---|
| OBJECTS | GS-ST | - | $0.78_{\pm 0.01}$ | $0.93_{\pm 0.06}$ | $0.21_{\pm 0.02}$ | $1.04_{\pm 0.03}$ |
| | REINFORCE | - | $0.51_{\pm 0.21}$ | $0.47_{\pm 0.12}$ | $0.14_{\pm 0.07}$ | $2.21_{\pm 1.28}$ |
| | VQEL-SP$_S$+MP | VQ-ST | $\mathbf{0.86_{\pm 0.01}}$ | $1.00_{\pm 0.00}$ | $0.19_{\pm 0.01}$ | $0.12_{\pm 0.02}$ |
| | GS-SP$_S$+MP | GS | $0.77_{\pm 0.01}$ | $1.00_{\pm 0.00}$ | $0.24_{\pm 0.03}$ | $1.15_{\pm 0.18}$ |
| | GS-SP$_S$+MP | GS-ST | $0.74_{\pm 0.06}$ | $1.00_{\pm 0.00}$ | $0.22_{\pm 0.03}$ | $1.25_{\pm 0.25}$ |
| SHAPE | GS-ST | - | $0.82_{\pm 0.01}$ | $1.00_{\pm 0.00}$ | $0.02_{\pm 0.01}$ | $1.21_{\pm 0.11}$ |
| | REINFORCE | - | $0.86_{\pm 0.00}$ | $0.83_{\pm 0.15}$ | $0.01_{\pm 0.01}$ | $0.88_{\pm 0.11}$ |
| | VQEL-SP$_S$+MP | VQ-ST | $\mathbf{0.91_{\pm 0.01}}$ | $1.00_{\pm 0.00}$ | $0.05_{\pm 0.02}$ | $0.39_{\pm 0.10}$ |
| | GS-SP$_S$+MP | GS | $0.81_{\pm 0.01}$ | $1.00_{\pm 0.00}$ | $0.05_{\pm 0.01}$ | $1.02_{\pm 0.09}$ |
| | GS-SP$_S$+MP | GS-ST | $0.82_{\pm 0.02}$ | $0.93_{\pm 0.12}$ | $0.04_{\pm 0.02}$ | $1.16_{\pm 0.09}$ |
| DSPRITES | GS-ST | - | $0.81_{\pm 0.01}$ | $0.90_{\pm 0.00}$ | $0.10_{\pm 0.01}$ | $1.80_{\pm 0.05}$ |
| | REINFORCE | - | $0.88_{\pm 0.02}$ | $0.80_{\pm 0.10}$ | $0.06_{\pm 0.00}$ | $1.06_{\pm 0.13}$ |
| | VQEL-SP$_S$+MP | VQ-ST | $\mathbf{0.93_{\pm 0.01}}$ | $1.00_{\pm 0.00}$ | $0.07_{\pm 0.01}$ | $0.40_{\pm 0.01}$ |
| | GS-SP$_S$+MP | GS | $0.79_{\pm 0.00}$ | $1.00_{\pm 0.00}$ | $0.11_{\pm 0.01}$ | $1.81_{\pm 0.04}$ |
| | GS-SP$_S$+MP | GS-ST | $0.77_{\pm 0.04}$ | $0.73_{\pm 0.23}$ | $0.10_{\pm 0.00}$ | $1.99_{\pm 0.32}$ |
| CELEBA | GS-ST | - | $0.90_{\pm 0.00}$ | $1.00_{\pm 0.00}$ | $0.14_{\pm 0.01}$ | $1.01_{\pm 0.08}$ |
| | REINFORCE | - | $\mathbf{0.93_{\pm 0.01}}$ | $1.00_{\pm 0.00}$ | $0.11_{\pm 0.03}$ | $0.90_{\pm 0.06}$ |
| | VQEL-SP$_S$+MP | VQ-ST | $\underline{0.91_{\pm 0.01}}$ | $1.00_{\pm 0.00}$ | $0.10_{\pm 0.04}$ | $0.57_{\pm 0.09}$ |
| | GS-SP$_S$+MP | GS | $0.90_{\pm 0.01}$ | $1.00_{\pm 0.00}$ | $0.15_{\pm 0.01}$ | $0.27_{\pm 0.02}$ |
| | GS-SP$_S$+MP | GS-ST | $0.88_{\pm 0.00}$ | $1.00_{\pm 0.00}$ | $0.18_{\pm 0.03}$ | $0.38_{\pm 0.07}$ |

Table 4: Comparison of GS with self-play and VQEL in the sender self-play setting. Despite using the same self-play protocol, VQEL consistently achieves higher accuracy than GS with self-play, highlighting the importance of the VQ module in enabling effective self-play.

## 5.5 Robustness to Communication Capacity

To evaluate whether the effectiveness of VQEL depends on a particular communication capacity, we investigate its performance under channels with varying vocabulary sizes and message lengths. Specifically, we conduct sender self-play experiments on ShapeWorld using five vocabulary sizes (10, 20, 30, 50, and 100) and three message lengths (4, 8, and 12).

Figure 7 reports the resulting test accuracies. Across all communication settings, VQEL consistently outperforms both GS-ST and REINFORCE, demonstrating that its performance gains are not tied to a specific vocabulary size or message length. Instead, VQEL remains effective over a broad range of channel capacities.

Figures 8 and 9 further analyze the communication behavior by reporting the AW and the number of unique messages. For both GS-ST and REINFORCE, increasing the vocabulary size leads to a decrease in AW, indicating that the larger vocabulary is not effectively utilized. In contrast, VQEL maintains an AW of 1 across all settings, suggesting that it continues to exploit the available vocabulary efficiently even when the vocabulary size is as large as 100. Consistent with this observation, VQEL also produces substantially more unique messages than both baselines across all communication capacities, indicating a richer and more expressive communication protocol.

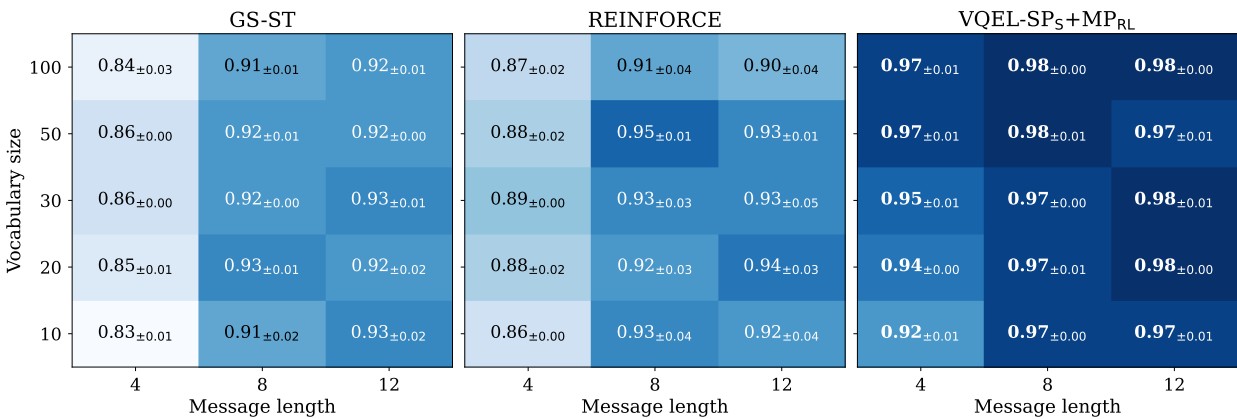

Figure 7: **Effect of communication capacity on test accuracy.** Test accuracy on ShapeWorld under different vocabulary sizes (10, 20, 30, 50, and 100) and message lengths (4, 8, and 12). VQEL consistently outperforms GS-ST and REINFORCE across all communication capacities, demonstrating robustness to both vocabulary size and message length.

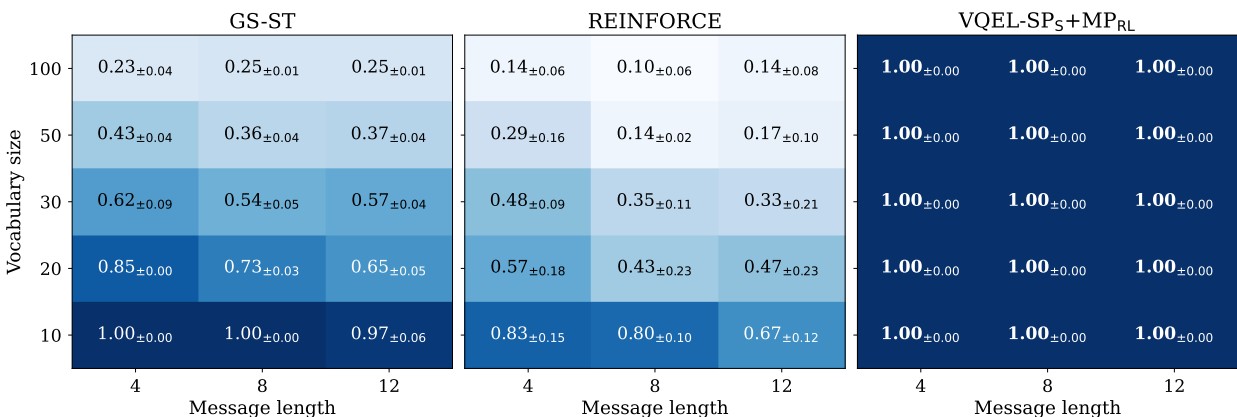

Figure 8: **Effect of communication capacity on Active Words (AW).** While the AW of GS-ST and REINFORCE decreases as the vocabulary size increases, VQEL maintains an AW of 1 across all settings, indicating effective utilization of the available vocabulary regardless of communication capacity.

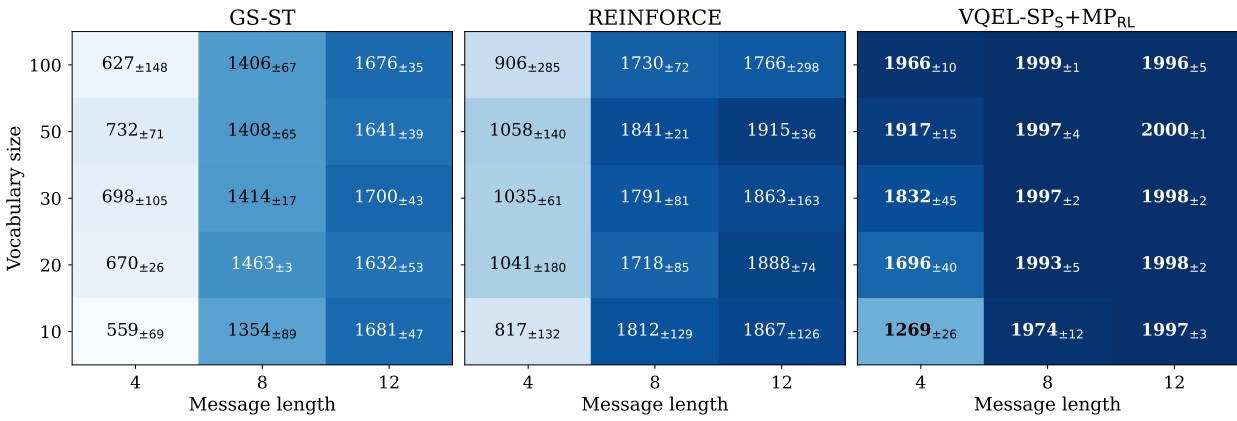

Figure 9: **Effect of communication capacity on number of unique messages.** VQEL consistently produces more unique messages than GS-ST and REINFORCE across all communication capacities, indicating a richer and more expressive communication protocol.

# 6 Conclusion

In this work, we address the optimization challenges of discrete communication by proposing Vector Quantized Emergent Language (VQEL). Inspired by the cognitive role of "inner speech," VQEL employs Vector Quantization to bootstrap language through differentiable self-play, effectively bridging private cognition and social communication. This approach allows agents to optimize internal representations via a learned codebook, providing a robust initialization for subsequent multi-agent interaction.

Our empirical evaluation across four diverse datasets—Synthetic Objects, ShapeWorld, dSprites, and CelebA, demonstrates the efficacy of this approach. We observed that:

- **Performance and Stability:** Agents pre-trained with VQEL self-play consistently outperform or match strong baselines (REINFORCE and Gumbel-Softmax) in terms of communication accuracy. VQEL exhibits superior stability, particularly as the number of distractors increases.

- **Vocabulary Efficiency:** Unlike baseline methods, which often suffer from vocabulary collapse, VQEL utilizes the available channel capacity fully (100% active words) and achieves lower entropy in concept-message mapping, indicating more precise communication.

- **The Necessity of Self-Play:** Our ablation studies confirm that the performance gains are not merely due to the VQ architecture but are driven by the self-play phase. Removing the self-play pre-training significantly degrades task success, validating the hypothesis that intrapersonal concept stabilization is a precursor to effective interpersonal communication.

These findings suggest that self-play provides a smoother optimization landscape for emergent language than trying to learn discrete protocols from scratch in a multi-agent setting.

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

## A   Euclidean Distance in Codebook

Table 5 reports the results of the sender self-play game when Euclidean distance is used in the codebook to select the nearest embedding vector. Across datasets, this choice leads to a 2–6% drop in accuracy compared to cosine similarity.

## B   Receiver Self-Play

In this experiment, the receiver first invents its own language during the self-play phase and then communicates with the sender in the mutual-play phase. During the mutual-play, the receiver can either be fine-tuned or frozen. As shown in Table 6, VQEL performs worse in both modes compared to the baselines and to VQEL in the previous experiments (Sender Self-Play and Sender and Receiver Self-Play). In particular, freezing the receiver results in lower accuracy than fine-tuning, indicating that the agent is unable to effectively transfer its language as a receiver.

This outcome is a direct consequence of how our method works, and it is precisely what our mechanism predicts. Successful language invention requires learning an effective codebook — the discrete vocabulary used for communication. In the mutual-play phase, this codebook is determined by the sender, since it is the sender that produces the symbolic message. When the receiver alone undergoes self-play, the codebook it develops during self-play is left unused: the sender must still learn its own codebook from scratch during mutual play. Consequently, receiver self-play confers no transferable benefit. By contrast, in the Sender Self-Play scenario, the codebook is already learned during self-play and is directly reused in mutual play, making optimization substantially easier.

We emphasize that this result does not contradict our central thesis — it confirms it. Our claim is that the benefit of self-play arises specifically because the symbol-producing agent (the sender) learns a usable codebook end-to-end prior to communication. Receiver self-play therefore serves as an ideal control for this hypothesis: by isolating self-play on the agent that does not own the codebook, it removes precisely the mechanism we posit to be responsible for the gains, and performance degrades accordingly. If self-play were merely a generic form of pretraining that improves performance regardless of role, it should help the receiver as well; the fact that it does not is exactly what our account predicts. The resulting asymmetry — producer-side self-play helps, receiver-side self-play does not — is thus a direct prediction of our mechanism rather than a counterexample to it, and provides further evidence that codebook learning by the sender is the source of VQEL's advantage.

| Dataset | Method | Sender Update | ACC ↑ | AW ↑ | TopSim ↑ | H(C\|M) ↓ |
|---------|--------|---------------|-------|------|----------|-----------|
| OBJECTS | GS-ST | - | $0.78_{\pm 0.01}$ | $0.93_{\pm 0.06}$ | $0.21_{\pm 0.02}$ | $1.04_{\pm 0.03}$ |
| | REINFORCE | - | $0.51_{\pm 0.21}$ | $0.47_{\pm 0.12}$ | $0.14_{\pm 0.07}$ | $2.21_{\pm 1.28}$ |
| | VQEL-SP$_S$ | - | $0.75_{\pm 0.02}$ | $1.00_{\pm 0.00}$ | $0.17_{\pm 0.02}$ | $0.30_{\pm 0.03}$ |
| | VQEL-SP$_S$+MP | Frozen | $0.81_{\pm 0.02}$ | $1.00_{\pm 0.00}$ | $0.17_{\pm 0.02}$ | $0.30_{\pm 0.03}$ |
| | VQEL-SP$_S$+MP | RL | $0.84_{\pm 0.01}$ | $1.00_{\pm 0.00}$ | $0.17_{\pm 0.02}$ | $0.29_{\pm 0.03}$ |
| | VQEL-SP$_S$+MP | RL+Pres | $0.84_{\pm 0.01}$ | $1.00_{\pm 0.00}$ | $0.18_{\pm 0.01}$ | $0.28_{\pm 0.03}$ |
| SHAPE | GS-ST | - | $0.82_{\pm 0.01}$ | $1.00_{\pm 0.00}$ | $0.02_{\pm 0.01}$ | $1.21_{\pm 0.11}$ |
| | REINFORCE | - | $0.86_{\pm 0.00}$ | $0.83_{\pm 0.15}$ | $0.01_{\pm 0.01}$ | $0.88_{\pm 0.11}$ |
| | VQEL-SP$_S$ | - | $0.77_{\pm 0.02}$ | $1.00_{\pm 0.00}$ | $0.05_{\pm 0.01}$ | $0.64_{\pm 0.07}$ |
| | VQEL-SP$_S$+MP | Frozen | $0.85_{\pm 0.02}$ | $1.00_{\pm 0.00}$ | $0.05_{\pm 0.01}$ | $0.64_{\pm 0.07}$ |
| | VQEL-SP$_S$+MP | RL | $0.88_{\pm 0.01}$ | $1.00_{\pm 0.00}$ | $0.05_{\pm 0.01}$ | $0.64_{\pm 0.07}$ |
| | VQEL-SP$_S$+MP | RL+Pres | $0.88_{\pm 0.01}$ | $1.00_{\pm 0.00}$ | $0.05_{\pm 0.01}$ | $0.63_{\pm 0.09}$ |
| DSPRITES | GS-ST | - | $0.81_{\pm 0.01}$ | $0.90_{\pm 0.00}$ | $0.10_{\pm 0.01}$ | $1.80_{\pm 0.05}$ |
| | REINFORCE | - | $0.88_{\pm 0.02}$ | $0.80_{\pm 0.10}$ | $0.06_{\pm 0.00}$ | $1.06_{\pm 0.13}$ |
| | VQEL-SP$_S$ | - | $0.78_{\pm 0.02}$ | $1.00_{\pm 0.00}$ | $0.09_{\pm 0.01}$ | $0.85_{\pm 0.09}$ |
| | VQEL-SP$_S$+MP | Frozen | $0.86_{\pm 0.01}$ | $1.00_{\pm 0.00}$ | $0.09_{\pm 0.01}$ | $0.85_{\pm 0.09}$ |
| | VQEL-SP$_S$+MP | RL | $0.87_{\pm 0.01}$ | $1.00_{\pm 0.00}$ | $0.09_{\pm 0.01}$ | $0.85_{\pm 0.11}$ |
| | VQEL-SP$_S$+MP | RL+Pres | $0.87_{\pm 0.01}$ | $1.00_{\pm 0.00}$ | $0.09_{\pm 0.01}$ | $0.82_{\pm 0.07}$ |
| CELEBA | GS-ST | - | $0.90_{\pm 0.00}$ | $1.00_{\pm 0.00}$ | $0.14_{\pm 0.01}$ | $1.01_{\pm 0.08}$ |
| | REINFORCE | - | $0.93_{\pm 0.01}$ | $1.00_{\pm 0.00}$ | $0.11_{\pm 0.03}$ | $0.90_{\pm 0.06}$ |
| | VQEL-SP$_S$ | - | $0.82_{\pm 0.02}$ | $1.00_{\pm 0.00}$ | $0.10_{\pm 0.01}$ | $0.76_{\pm 0.01}$ |
| | VQEL-SP$_S$+MP | Frozen | $0.88_{\pm 0.01}$ | $1.00_{\pm 0.00}$ | $0.10_{\pm 0.01}$ | $0.76_{\pm 0.01}$ |
| | VQEL-SP$_S$+MP | RL | $0.89_{\pm 0.01}$ | $1.00_{\pm 0.00}$ | $0.10_{\pm 0.02}$ | $0.73_{\pm 0.03}$ |
| | VQEL-SP$_S$+MP | RL+Pres | $0.91_{\pm 0.00}$ | $1.00_{\pm 0.00}$ | $0.10_{\pm 0.00}$ | $0.58_{\pm 0.02}$ |

Table 5: Performance comparison across datasets and evaluation metrics for the sender self-play game using Euclidean distance in the codebook.

## C   Analysis of Message Compositionality

To investigate the degree of compositional structure learned by our method compared to baseline approaches, we analyze the distribution of message tokens conditioned on input attributes in SHAPEWORLD, specifically *shape* and *color*.

Figures 10, 11, and 12 report token distributions across message positions for VQEL (with sender self-play), REINFORCE, and GS-ST, respectively.

Across both baseline methods, the conditional token distributions appear relatively diffuse, with no clear alignment between specific tokens at fixed positions and semantic attributes. In other words, token usage is largely distributed in a manner that does not reveal strong token-wise specialization for shape or color information.

In contrast, VQEL exhibits more structured and selective token usage. Certain tokens become strongly associated with specific attributes. For instance, when the color is *blue*, at position 2 token 4 is produced with notably higher probability (approximately 0.4). Similarly, when the shape is *rectangle*, at position 1 token 2 shows a strong preference (approximately 0.35).

These patterns suggest that VQEL induces a more interpretable and partially disentangled communication protocol, where specific message components can be more directly associated with underlying semantic factors.

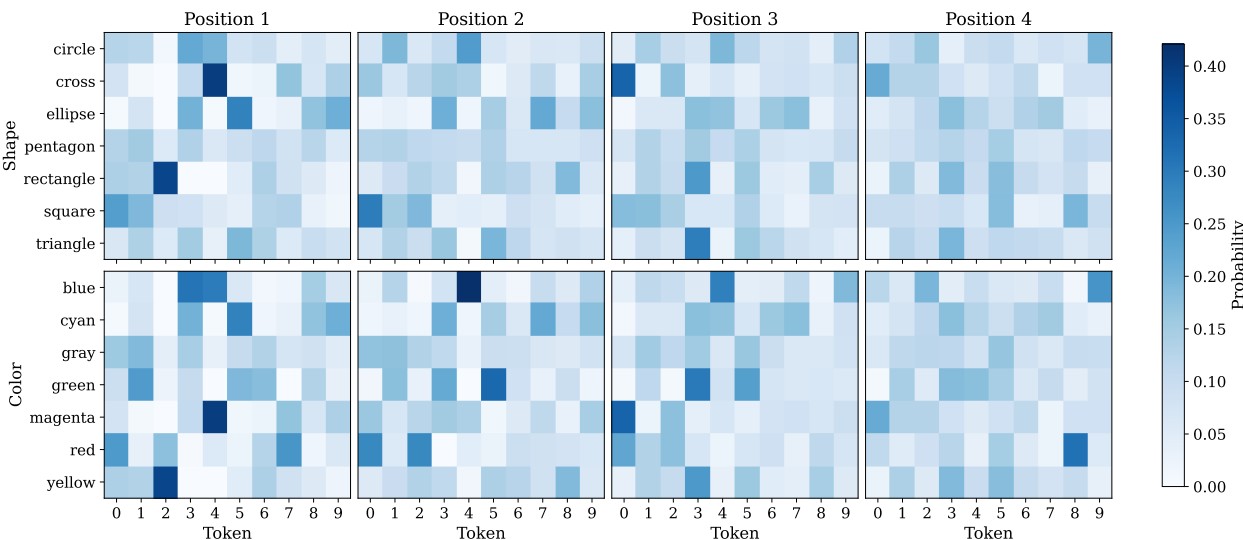

Figure 10: Token distributions conditioned on *shape* and *color* attributes for messages generated by VQEL with sender self-play. Each heatmap shows $P(\text{token} \mid \text{attribute value}, \text{position})$, where rows correspond to attribute values and columns correspond to message tokens.

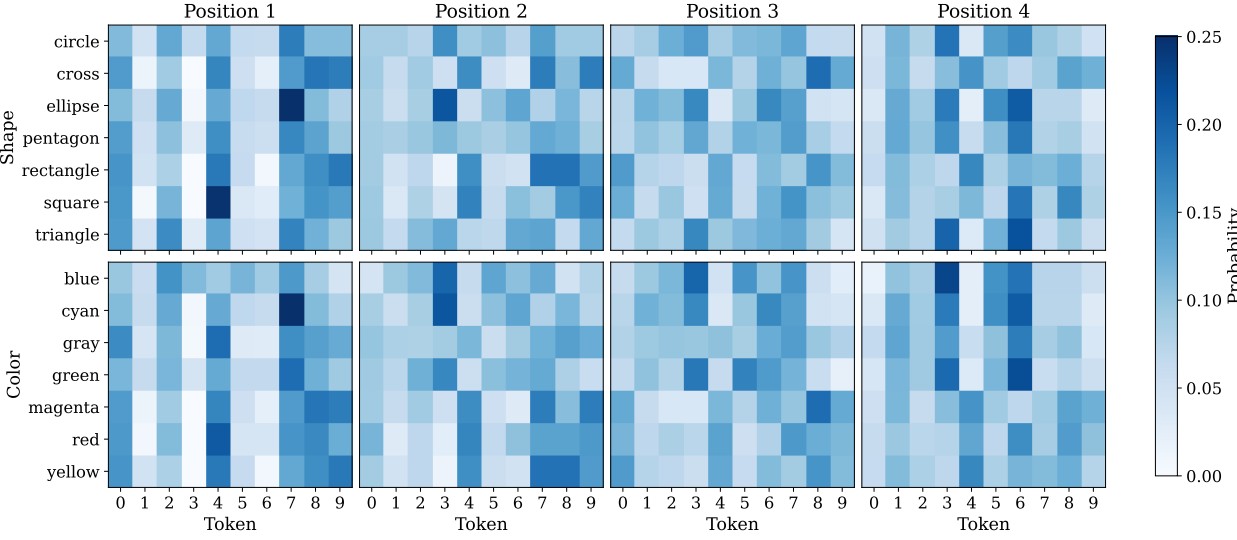

Figure 11: Token distributions conditioned on *shape* and *color* attributes for messages generated by RE-INFORCE. Each heatmap shows $P(\text{token} \mid \text{attribute value}, \text{position})$, where rows correspond to attribute values and columns correspond to message tokens.

| Dataset | Method | Receiver Update | ACC ↑ | AW ↑ | TopSim ↑ | H(C\|M) ↓ |
|---------|--------|-----------------|-------|------|----------|-----------|
| OBJECTS | GS-ST | - | $0.78_{\pm0.01}$ | $0.93_{\pm0.06}$ | $0.21_{\pm0.02}$ | $1.04_{\pm0.03}$ |
| | REINFORCE | - | $0.51_{\pm0.21}$ | $0.47_{\pm0.12}$ | $0.14_{\pm0.07}$ | $2.21_{\pm1.28}$ |
| | VQEL-SP$_R$ | - | $0.82_{\pm0.01}$ | $1.00_{\pm0.00}$ | $0.19_{\pm0.01}$ | $0.12_{\pm0.02}$ |
| | VQEL-SP$_R$+MP | Frozen | $0.14_{\pm0.02}$ | $0.70_{\pm0.10}$ | $0.14_{\pm0.03}$ | $3.97_{\pm0.26}$ |
| | VQEL-SP$_R$+MP | Fine-tuned | $0.43_{\pm0.12}$ | $1.00_{\pm0.00}$ | $0.18_{\pm0.02}$ | $2.17_{\pm0.40}$ |
| SHAPE | GS-ST | - | $0.82_{\pm0.01}$ | $1.00_{\pm0.00}$ | $0.02_{\pm0.01}$ | $1.21_{\pm0.11}$ |
| | REINFORCE | - | $0.86_{\pm0.00}$ | $0.83_{\pm0.15}$ | $0.01_{\pm0.01}$ | $0.88_{\pm0.11}$ |
| | VQEL-SP$_R$ | - | $0.87_{\pm0.01}$ | $1.00_{\pm0.00}$ | $0.05_{\pm0.03}$ | $0.38_{\pm0.10}$ |
| | VQEL-SP$_R$+MP | Frozen | $0.40_{\pm0.16}$ | $0.83_{\pm0.12}$ | $0.03_{\pm0.00}$ | $2.04_{\pm0.10}$ |
| | VQEL-SP$_R$+MP | Fine-tuned | $0.85_{\pm0.01}$ | $1.00_{\pm0.00}$ | $0.02_{\pm0.01}$ | $0.65_{\pm0.05}$ |
| DSPRITES | GS-ST | - | $0.81_{\pm0.01}$ | $0.90_{\pm0.00}$ | $0.10_{\pm0.01}$ | $1.80_{\pm0.05}$ |
| | REINFORCE | - | $0.88_{\pm0.02}$ | $0.80_{\pm0.10}$ | $0.06_{\pm0.00}$ | $1.06_{\pm0.13}$ |
| | VQEL-SP$_R$ | - | $0.91_{\pm0.02}$ | $1.00_{\pm0.00}$ | $0.07_{\pm0.01}$ | $0.40_{\pm0.02}$ |
| | VQEL-SP$_R$+MP | Frozen | $0.24_{\pm0.09}$ | $0.67_{\pm0.06}$ | $0.09_{\pm0.01}$ | $4.56_{\pm0.52}$ |
| | VQEL-SP$_R$+MP | Fine-tuned | $0.86_{\pm0.01}$ | $1.00_{\pm0.00}$ | $0.07_{\pm0.00}$ | $0.80_{\pm0.12}$ |
| CELEBA | GS-ST | - | $0.90_{\pm0.00}$ | $1.00_{\pm0.00}$ | $0.14_{\pm0.01}$ | $1.01_{\pm0.08}$ |
| | REINFORCE | - | $0.93_{\pm0.01}$ | $1.00_{\pm0.00}$ | $0.11_{\pm0.03}$ | $0.90_{\pm0.06}$ |
| | VQEL-SP$_R$ | - | $0.89_{\pm0.01}$ | $1.00_{\pm0.00}$ | $0.10_{\pm0.04}$ | $0.58_{\pm0.10}$ |
| | VQEL-SP$_R$+MP | Frozen | $0.40_{\pm0.04}$ | $0.77_{\pm0.23}$ | $0.12_{\pm0.06}$ | $3.80_{\pm0.64}$ |
| | VQEL-SP$_R$+MP | Fine-tuned | $0.54_{\pm0.05}$ | $1.00_{\pm0.00}$ | $0.12_{\pm0.01}$ | $2.48_{\pm0.24}$ |

Table 6: Performance comparison across datasets and evaluation metrics for the receiver self-play game.

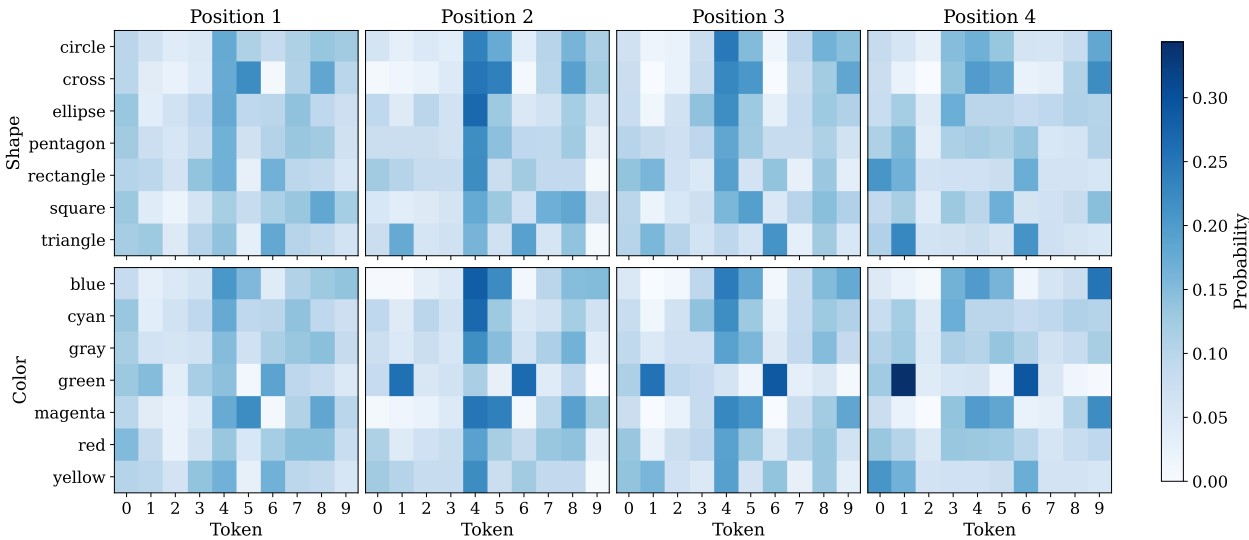

Figure 12: Token distributions conditioned on *shape* and *color* attributes for messages generated by GS-ST. Each heatmap shows $P(\text{token} \mid \text{attribute value}, \text{position})$, where rows correspond to attribute values and columns correspond to message tokens.

