# OpenReview forum: "VQEL: Enabling Self-Play in Emergent Language Games via Agent Internal Vector Quantization"
_TMLR — Under review for TMLR_

### Review · Reviewer_Q2hW · 2026-05-14

**Summary Of Contributions:**

The paper proposes VQEL (Vector Quantized Emergent Language), a method for single-turn referential games. A VQ-VAE style codebook is inserted into the sender's message generation module so that, during a self-play pretraining phase, the sender can act as its own receiver and be trained end-to-end via the straight-through estimator and a CLIP-style contrastive loss. After self-play, the agent enters mutual play with a separate agent, where the sender is fine-tuned with REINFORCE while the receiver is trained by backpropagation. Experiments are conducted on four datasets (Synthetic Objects, ShapeWorld, dSprites, CelebA) against REINFORCE and Gumbel-Softmax (ST) baselines. The authors report that VQEL improves task accuracy, achieves 100% active vocabulary use, lowers H(C|M), and is more robust to increasing distractor counts. An ablation removing self-play (VQEL-MP) attributes most of the gain to the self-play phase rather than to the VQ module alone.

Key strengths: a clean motivation, a fairly complete set of self-play ablations (sender-only, receiver-only, sender+receiver, no-self-play), and clear vocabulary efficiency results.

Key weaknesses: the methodological novelty is limited to placing a standard VQ-VAE bottleneck inside a standard signaling-game architecture; the most directly related prior work (Carmeli et al. 2023, "Emergent Quantized Communication") is mentioned but never used as a baseline, and baselines are otherwise restricted to REINFORCE (1992) and Gumbel-Softmax (2017); all evaluation is in-distribution on small visual referential games despite the broad title; the Receiver Self-Play results in Appendix B contradict the central claim and are not reconciled; many "best" entries in Tables 1 and 2 are within one standard deviation of competing entries.

**Audience:**

No

**Audience Explanation:**

First, the claim that VQEL is a principled solution to discrete emergent communication rests on a comparison set restricted to REINFORCE (1992) and Gumbel-Softmax ST (2017), which excludes the most directly competing approach. Carmeli et al. 2023 ("Emergent Quantized Communication") proposes a quantization-based emergent communication scheme that is acknowledged in Section 2.1 but never used as a baseline. Kharitonov et al. 2020 ("Entropy Minimization in Emergent Languages") proposes another stabilization scheme for the same discrete optimization problem and is also absent from the comparison. Without these baselines, the reported improvements cannot be attributed to the proposed self-play mechanism rather than to known stabilization effects of any VQ-style discrete bottleneck.

Second, the framing as "Emergent Language Games" is not matched by the experimental scope. All experiments use single-turn signaling games on small in-distribution visual datasets with a vocabulary size of 10 and message length 4. ShapeWorld is described as having a compositional train/test split, but no compositional or out-of-distribution accuracy is actually reported. TopSim values stay very low (0.02 to 0.21) across all methods, indicating that no method, including VQEL, produces a structured or compositional language under this setup.

Third, the central claim that intrapersonal self-play is a useful precursor to communication is contradicted by the Receiver Self-Play results in Appendix B, where VQEL drops to ACC values between 0.14 and 0.54 in several settings and is consistently below the baselines. The paper acknowledges this but does not reconcile it with the main thesis.

Fourth, several bolded entries in Tables 1 and 2 differ from competing entries by less than the reported standard deviation, so some of the headline improvements may not be statistically meaningful.

Finally, the practical value of the experimental setup itself is questionable in the current research context. Stripped of the "emergent language" framing, the task reduces to a contrastive learning problem in which a sender encodes a target image into a short discrete code that a receiver uses to retrieve the target from a small candidate pool. In 2026, with the rapid progress of large language models and multimodal foundation models, it is unclear what method design insights, practical utility, or interpretability gains this setup can offer to frontier multimodal retrieval. The paper does not establish a connection to any modern multimodal retrieval pipeline, does not show that the learned codebook indices transfer to any task involving real language or larger-scale data, and does not argue why a 10-word vocabulary on toy images should be informative beyond the narrow EL community.

**Broader Impact Concerns:**

None of ethical concerns.

**Claims And Evidence:**

No

**Claims Explanation:**

I think the claims made in this  submission are partially supported.

**Requested Changes:**

Please refer to the last section.

---

> ### Author Response · Authors · 2026-06-28
>
> We sincerely thank Reviewer Q2hW for the rigorous critique and detailed feedback. We appreciate the critical perspective, as it has pushed us to clarify our scope, refine our claims, and add important new experiments to the manuscript. Based on your review, we have significantly narrowed our overarching claims to position VQEL strictly as a principled optimization strategy, explicitly clarified the Out-of-Distribution (OOD) nature of our ShapeWorld evaluation, added new robustness experiments regarding channel capacities (Section 5.5), and restructured our discussion on Receiver Self-Play.
>
> Below, we address your concerns point by point:
>
> **Response to Point 1 (Missing Baselines and Attribution - VQ vs. Self-play):**
> We address the three sub-points—Carmeli et al. (2023), Kharitonov et al. (2020), and the concern about attribution—in turn:
> * **Carmeli et al. (2023):** We discuss this work in Section 2.1, but it is not an appropriate direct baseline because its setting differs fundamentally from ours. First, their method uses uniform scalar quantization (simple numerical rounding independent for each dimension) rather than learning a semantic, vector-quantized codebook. Second, their game is played over a *continuous* communication channel during training. What is transmitted is a continuous vector, not a discrete symbol. Our setting addresses the strictly symbolic, discrete channel, which is the core non-differentiable bottleneck in EL.
> * **Kharitonov et al. (2020):** We believe there is a misunderstanding regarding the nature of this work. It does not propose a "stabilization scheme" or a training algorithm for the non-differentiability problem. Rather, it is an analysis paper studying an information-theoretic property of emergent languages (entropy-minimization pressure). There is no standalone "method" from this paper to run as a baseline against VQEL.
> * **Attribution (Self-play vs. any VQ bottleneck):** We completely agree this is a key concern, which is why we addressed it directly in our ablation study (**Section 5.3, Table 4**). In the "VQEL-MP" setting, we trained the model using the VQ bottleneck but *without* the self-play phase. This setting performs dramatically worse than full VQEL (e.g., on Objects: 0.86 with self-play vs. 0.28 without; on DSprites: 0.93 vs. 0.88). Without the self-play phase to stabilize the codebook, the VQ-style discrete bottleneck fails and is often worse than standard REINFORCE. The performance gains are therefore attributable specifically to the self-play mechanism.
>
> **Response to Point 2 (Framing, Experimental Scope, and TopSim):**
> We have explicitly reframed VQEL not as a model of full human language emergence, but as a principled optimization and initialization strategy. However, we respectfully stand by our experimental scope:
> * **Validity of the single-turn game:** The single-turn Lewis signaling game is the foundational testbed and gold standard in emergent communication research (e.g., *Lazaridou et al., 2017; Havrylov & Titov, 2017; Chaabouni et al., 2020*). Evaluating across diverse datasets on this benchmark is a requirement to isolate and study the fundamental optimization bottleneck of non-differentiable channels.
> * **Channel Capacity (New Experiments):** To address your concern about the $V=10$ and $L=4$ bottleneck, we conducted extensive new experiments varying vocabulary sizes (up to 100) and message lengths. As detailed in the new **Section 5.5**, VQEL consistently maintains its optimization benefits and achieves 100% vocabulary utilization across all capacities, whereas baselines suffer from vocabulary collapse.
> * **ShapeWorld and OOD Generalization:** We apologize for the lack of clarity. In our ShapeWorld setup, the test set consists *entirely* of held-out, unseen color-shape combinations. Thus, the standard test accuracy reported in Table 1 is, by definition, the **compositional Out-Of-Distribution (OOD) accuracy**. We have updated **Section 4.1** to make this absolutely explicit.
> * **Low TopSim:** As extensively documented in the literature (*Yao et al., 2022; Chaabouni et al., 2020*), TopSim does not strictly correlate with task success. Our claim is that VQEL provides a superior optimization mechanism leading to higher accuracy, perfect channel utilization, and training stability. Low TopSim does not contradict these claims. Furthermore, our new token-level analysis in **Appendix C** shows that VQEL produces a much more structured and interpretable protocol than the baselines.
> Our claim is that VQEL provides a superior optimization mechanism that leads to:
> Higher and more stable task success (Accuracy).
> Perfect utilization of the channel capacity, avoiding the vocabulary collapse seen in baselines.
> More efficient and certain mappings (lower $H(C|M)$).
> Low TopSim does not contradict any of these claims. VQEL successfully solves the variance and stability issues of discrete channels without degrading compositionality compared to the baselines.

---

> ### Author Response · Authors · 2026-06-28
>
> **Response to Point 3 (Receiver Self-Play - A Confirming Control, Not a Contradiction):**
>
> We thank the reviewer for raising this point, and we respectfully disagree that the Receiver Self-Play results contradict our central thesis — in fact, they confirm it.
>
> Our claim is not that self-play is beneficial for any agent regardless of its role. Rather, it is that the benefit of self-play arises specifically because the symbol-producing agent (the sender) learns a usable codebook end-to-end prior to communication. The codebook — the discrete vocabulary used for communication — is constructed by the sender, since it is the sender that produces the symbolic message during mutual play. When the receiver alone undergoes self-play, it does not build this codebook; consequently, its self-play phase confers no transferable benefit, and the sender is still forced to learn a codebook from scratch during mutual play. The low accuracies the reviewer notes are precisely the expected consequence of this.
>
> We would emphasize that this makes Receiver Self-Play an ideal control for our hypothesis rather than a counterexample to it. By isolating self-play on the agent that does not own the codebook, it removes the very mechanism we posit to be responsible for the gains, and performance degrades accordingly. The resulting asymmetry — producer-side self-play helps, receiver-side self-play does not — is a direct prediction of our mechanism. If self-play were merely a generic form of pretraining that improves performance regardless of role, it should help the receiver as well; the fact that it does not is exactly what our account predicts, and provides further evidence that codebook learning by the sender is the source of VQEL's advantage.
>
> To address the reviewer's concern that this was acknowledged but not reconciled with the main thesis, we have revised the main text (the Receiver Self-Play paragraph in **Section 4.2** and **Appendix B**) to explicitly frame this result as a confirming control for our mechanism rather than as an unexplained limitation.
>
> **Response to Point 4 (Statistical Significance - Within 1-std margins):**
>
> We would like to clarify an important point: the entries that fall within one standard deviation of each other are the **internal VQEL variants** (Frozen / RL / RL+Pres), not VQEL versus the baselines. These variants are expected to perform similarly, since they share the same self-play–pretrained codebook and differ only in how the sender is updated during mutual play; the comparison among them is not a headline claim of the paper.
>
> The central claim — that VQEL improves over the REINFORCE and GS-ST baselines — holds with a margin well beyond one standard deviation in nearly all settings. For example, on Objects, VQEL reaches $0.86$ versus $0.78$ for GS-ST and $0.51 \pm 0.21$ for REINFORCE; on DSprites, $0.93$ versus $0.81$ for GS-ST; and on Shape, $0.91$ versus $0.82$ for GS-ST. Thus, the improvements that constitute the paper's main message are statistically meaningful, and the within-std differences the reviewer refers to concern only the relative ranking among VQEL's own variants, which does not affect our conclusions.

---

> ### Author Response · Authors · 2026-06-28
>
> **Response to Point 5 (Practical Value and Framing vs. Multimodal Retrieval):**
> Emergent language is an active and well-established research area in which numerous papers are regularly published across a diverse set of subfields — ranging from referential and reconstruction games to question-answer and grid-world games — with distinct lines of inquiry into compositionality, interpretability, population dynamics, and evaluation metrics. Each year, a substantial number of papers on emergent language appear across a wide range of venues, including journals such as TMLR itself and top-tier conferences such as NeurIPS, ICLR, and ICML. The maturity and ongoing vitality of the field are further evidenced by recently published comprehensive surveys and taxonomies and dedicated reviews of its applications across machine learning, NLP, linguistics, and cognitive science.
>
> The field is important for several reasons. First, it offers a controlled computational framework for studying one of the deepest open questions in science — how language and communication arise — providing testable insights into the mechanisms of human language evolution and acquisition that are difficult to probe in natural settings. Second, as artificial agents are increasingly deployed in multi-agent systems, understanding how communication protocols emerge, stabilize, and can be aligned with human language is directly relevant to building interpretable, controllable, and trustworthy agentic systems.
>
> We respectfully note that the reviewer frames our contribution as a "multimodal retrieval" problem, but this is not the lens through which our work is positioned or should be evaluated: our work is explicitly framed as a study within emergent language, where the referential (Lewis signaling) game is the canonical, widely-adopted testbed for studying how discrete communication protocols arise among agents. Evaluating the contribution against the goals of frontier multimodal retrieval pipelines therefore applies a criterion that is orthogonal to the research question we address.
>
> ---
> **References:**
> [1] Lazaridou, A., et al. (2017). Multi-agent cooperation and the emergence of (natural) language. *ICLR*.
>
> [2] Havrylov, S., & Titov, I. (2017). Emergence of Language with Multi-agent Games: Learning to Communicate with Sequences of Symbols. *NeurIPS*.
>
> [3] Bouchacourt, D., & Baroni, M. (2018). How agents see things: On visual representations in an emergent language game. *EMNLP*.
>
> [4] Chaabouni, R., et al. (2020). Compositionality and Generalization in Emergent Languages. *ACL*.
>
> [5] Yao, X., et al. (2022). Linking Emergent and Natural Languages via Corpus Transfer. *ICLR*.

---

### Review · Reviewer_yijr · 2026-05-27

**Summary Of Contributions:**

The paper studies emergent communication with discrete symbols in a cooperative task. This faces a technical challenge: sampling discrete tokens is non-differentiable. The paper proposes a novel approach to solve this through VQ that discretizes continuous internal representations into a finite codebook of embedding vectors. Additionally, the paper proposes an addition of self-play to the architecture. VQEL achieves higher or comparable accuracy to the baselines on all datasets, is more robust to the number of the distractors, and utilizes the vocabulary more efficiently than the baselines.

Strengths:
1. The paper is clearly written and well-structured.
2. The analyses are thorough: multiple diverse datasets, multiple metrics, some key ablations in Table 3 (though some are missing, see Requested changes below).

Weaknesses:

1. The setup is very small-scale (vocabulary size of 10 and message length of 4) so it’s unclear whether the results would hold in a more realistic scenario.
2. The cognitive framing is weak — the authors draw an analogy between Fodor’s language of thought and self-play but the setup and evaluations have little relation to Fodor’s ideas.
3. Some ablations are missing (see Requested changes below).

**Audience:**

Yes

**Audience Explanation:**

Yes, emergent communication protocols between AI agents is a growing topic and the findings will be interesting to the TMLR community.

**Claims And Evidence:**

Yes

**Claims Explanation:**

The paper provides consistent evidence for the claims. In particular, the experiments are performed on multiple diverse datasets with multiple metrics reported. The authors report multiple runs with different seeds along with the standard deviations.

**Requested Changes:**

1. The approach is essentially 2 contributions: one is architectural — leveraging VQ for the problem — and one is through adding self-play. The ablations reported in Table 3 only compare VQEL+self-play and VQEL-self-play. I’m convinced that removing self-play results in a performance drop. But I’m not convinced that VQ provides a benefit beyond Gumbel-softmax pretrained via self-play.

2. I’d like to learn more about what the invented language looks like — does it have something resembling compositional structure? How does it change through mutual play? The authors do report the TopSim metric that speaks to this to some degree but it’s not very informative as the authors acknowledge. Have you looked at the performance on the held-out images in ShapeWorld for instance? Could one visualize the learned codebook in some form (e.g., t-SNE, etc)?

3. The first bullet point in the conclusions suggests that VQEL has superior performance and stability. I expected some analyses of training stability to support this. Could you provide the evidence to substantiate the stability claim?

4. REINFORCE and VQEL-MP (Table 3) have unexpectedly low accuracy with high variance on the Objects dataset. Do you have an intuition for what’s going on?

---

> ### Author Response · Authors · 2026-06-28
>
> We sincerely thank Reviewer yijr for the highly constructive feedback, thorough evaluation, and excellent suggestions. Your insightful prompts directly led to several new experiments and analyses—including the Gumbel-Softmax ablation, the stability learning curves, channel capacity scaling, and message compositionality visualizations—all of which have significantly strengthened the paper. Before addressing your specific requested changes, we would like to highlight how we addressed the two general weaknesses you accurately pointed out in your review:
>
> **A. Regarding the small-scale setup (Vocabulary and Message Length):**
> We completely agree that evaluating robustness to larger capacities is important. To address this weakness, we conducted extensive new experiments varying the vocabulary size (up to 100) and message length (up to 12). These results are now detailed in the new **Section 5.5 (Robustness to Communication Capacity)** and **Figures 7-9**. VQEL consistently maintains its optimization benefits and achieves 100% vocabulary utilization across all capacities, whereas the baselines suffer from severe vocabulary collapse when scaled up.
>
> **B. Regarding the cognitive framing:**
> We agree with your assessment. We have revised the Introduction (Section 1), and Conclusion to explicitly tone down the cognitive framing.
>
> Below, we address your specifically requested changes point by point:
>
> **Response to Point 1 (VQ vs. Gumbel-Softmax pretrained via self-play):**
>
> We sincerely thank the reviewer for pointing out this critical nuance. This is an excellent question: does the performance gain come strictly from the pretraining phase, or is the VQ architecture itself essential?
>
> To answer this explicitly, we designed exactly the experiment you suggested. We applied our identical self-play protocol to a sender utilizing Gumbel-Softmax (both standard GS and GS-ST) instead of VQ. The results of this experiment have been added to the revised manuscript in the new **Section 5.4 (Effect of Vector Quantization)** and are detailed in **Table 5**.
>
> As shown in Table 5, applying self-play to Gumbel-Softmax (`GS-SP_S+MP`) does not yield the same performance gains and consistently underperforms our proposed `VQEL-SP_S+MP`. In some cases, GS with self-play even performs worse than the baseline GS without self-play.
>
> *Why does VQ succeed where GS fails in self-play?*
> The core reason lies in how the two methods handle representations, which highlights that the synergy of VQ and self-play is fundamentally more than just "pretraining."
>
> Gumbel-Softmax is primarily a stochastic routing mechanism designed to pass gradients through categorical samples. It does not natively build a structured, persistent conceptual space. In contrast, the VQ module maintains an explicit codebook. This naturally provides two complementary representations: a continuous embedding for dense gradient flow during self-play, and a discrete index that translates seamlessly to a symbolic channel.
>
> Consequently, VQEL establishes a true discrete self-play mechanism. Rather than simply updating network weights (as generic pretraining does), the combination of VQ’s specific discrete structural bottleneck and the self-play objective forces the agent to invent a foundational, internal "proto-language". This structurally sound proto-language serves as an optimal starting point for interpersonal communication—something Gumbel-Softmax is architecturally incapable of achieving, even with self-play.
>
> Ultimately, our findings (Table 5) prove that the objective is not just to pretrain the network, but to utilize a discrete self-play mechanism that autonomously stabilizes concepts prior to multi-agent interaction. We believe this new ablation significantly strengthens the paper, and we are deeply grateful for your suggestion that led to it.

---

> ### Author Response · Authors · 2026-06-28
>
> **Response to Point 2 (Visualizing the language and ShapeWorld OOD performance):**
>
> We sincerely thank the reviewer for this excellent suggestion. It prompted us to provide a much deeper qualitative analysis of the learned protocols, which has significantly enriched the paper.
>
> * **1. Held-out performance in ShapeWorld:**
> We apologize for not making this explicit in the original manuscript. Our ShapeWorld test split consists *entirely* of held-out, unseen color-shape combinations. Therefore, the test accuracies we reported in our tables **already measure compositional, out-of-distribution (OOD) generalization**. To prevent any future misunderstandings, we have added the following clarification to **Section 4.1**:
> > *"Consequently, all reported ShapeWorld accuracies measure compositional, out-of-distribution generalization to unseen color–shape combinations."*
> The fact that VQEL consistently achieves high accuracy on this split demonstrates its strong compositional generalization capabilities.
>
> * **2. Visualizing the invented language and its compositional structure:**
> To directly address your question about what the language looks like, we added a new comprehensive analysis in **Appendix C (Analysis of Message Compositionality)**.
> Rather than plotting a t-SNE of the continuous embeddings (which only shows spatial geometry), we analyzed the actual discrete communication protocol. We plotted the conditional token distributions—$P(\text{token} \mid \text{attribute, position})$—for the ShapeWorld dataset (see **Figures 10, 11, and 12** in the revised manuscript).
> The results are highly revealing:
>   * **Baselines (REINFORCE & GS-ST):** As shown in Figures 11 and 12, the baseline methods exhibit relatively diffuse and entangled token distributions, meaning there is no clear, interpretable alignment between specific message tokens and semantic attributes.
>   * **VQEL:** As shown in Figure 10, VQEL develops a highly structured, selective, and partially disentangled (compositional) protocol. Specific tokens become strongly bound to specific attribute values at fixed positions. For instance, token 4 at position 2 is highly specific to the color *blue*, and token 2 at position 1 strongly encodes the shape *rectangle*.
>
> This visualization concretely confirms that the self-play phase in VQEL not only improves raw accuracy but actually induces a much more interpretable, compositionally structured language than standard RL or Gumbel-Softmax. We are very grateful for this prompt, as it allowed us to demonstrate this structural advantage clearly.
>
>
> **Response to Point 3 (Training Stability):**
>
> We completely agree with the reviewer that such a central claim requires explicit empirical substantiation. To address this, we have added a dedicated stability analysis to the revised manuscript.
>
> Specifically, we provided the following evidence:
> * **New Visual Evidence (Learning Curves):** We added a new figure (**Figure 6**) that plots the training accuracy across multiple random seeds throughout the training process. The shaded regions denote one standard deviation. As the figure clearly demonstrates, VQEL exhibits a significantly smoother learning trajectory and much narrower variance compared to the baselines (especially REINFORCE), preventing the severe performance collapses seen during training on the Objects and Shape datasets.
> * **Detailed Textual Analysis:** We expanded the discussion at the end of **Section 5.3** (Effect of Self-Play) to formally interpret these stability results. Furthermore, we explicitly analyzed the quantitative variance reported in Table 4. For instance, on the Objects dataset, learning from scratch with REINFORCE yields a highly unstable $0.51 \pm 0.21$ accuracy, and VQEL without self-play is similarly fragile ($0.28 \pm 0.18$). In stark contrast, VQEL with self-play achieves $0.86 \pm 0.01$, drastically reducing the standard deviation and confirming the robust optimization landscape provided by the pre-established codebook.
>
> We believe that this combination of learning curves and expanded variance analysis comprehensively substantiates the stability claims made in the conclusion. We thank the reviewer for prompting us to make this evidence explicit.

---

> ### Author Response · Authors · 2026-06-28
>
> **Response to Point 4 (Unstable/low accuracy on the Objects dataset):**
>
> Thank you for this question — we do have a clear intuition, and it directly supports our central thesis.
>
> The Objects dataset is the hardest setting for learning a protocol from scratch. Unlike the visual datasets, its inputs are discrete one-hot vectors (four attributes × ten values = 10,000 unique objects) with no gradual similarity structure between objects. The agent must therefore learn an essentially arbitrary mapping from a large discrete input space to a discrete protocol, which is precisely the regime where REINFORCE's high-variance, weak scalar reward struggles most. Depending on the random seed, optimization sometimes finds a reasonable protocol and sometimes collapses into a degenerate one, which is why REINFORCE shows both low mean accuracy and high variance ($0.51 \pm 0.21$) here.
>
> VQEL-MP (VQEL without self-play) is even more fragile ($0.28 \pm 0.18$) because it must simultaneously learn two difficult things from scratch — a meaningful codebook and the object-to-symbol mapping — using only the same high-variance REINFORCE signal through the discrete channel. Without the self-play phase to first stabilize the codebook, this joint optimization is harder than plain REINFORCE.
>
> Importantly, this is exactly the behavior our thesis predicts: the instability on Objects is a symptom of learning discrete protocols from scratch in a hard combinatorial space. When self-play is used, VQEL reaches $0.86 \pm 0.01$ on the same dataset — both high and stable — showing that the self-play phase is precisely what resolves this instability. We see this contrast as one of the clearest illustrations of why intrapersonal self-play provides a smoother optimization landscape than learning the protocol directly in mutual play.
>
> We have incorporated this explanation into the revised paper as an additional paragraph in the **Section 5.3 (Effect of Self-Play)**, where it accompanies the VQEL-MP ablation in Table 4 and explicitly references the relevant numbers, so that readers can directly connect the observed instability on Objects to the role of the self-play phase.

---

> > ### Comment · Reviewer_yijr · 2026-06-29
> >
> > Thank you for the detailed response and additional experiments. The new Gumbel-Softmax + self-play ablation, the larger-scale capacity experiments, the stability curves, along with the compositionality visualizations collectively address my concerns.

---

### Review · Reviewer_bh6g · 2026-06-15

**Summary Of Contributions:**

The paper proposes VQEL, a method for learning discrete communication between agents. The main idea is to let an agent first learn an internal discrete codebook through self-play, and then use this learned codebook as the basis for communication with another agent in mutual play. This is meant to make learning symbolic messages more stable than starting from scratch with REINFORCE or Gumbel-Softmax.
I see the main strength as the simple use of vector quantization to create a discrete message space before multi-agent communication. The experiments show improved or competitive accuracy on several datasets, and the ablation suggests that self-play is important. A weakness is that the paper sometimes describes the learned codes as “language” in a strong sense, while the experiments mainly show task-specific communication protocols.

**Additional Comments:**

I found the core idea interesting and the experimental direction worthwhile. My main recommendation is to present the contribution more narrowly and precisely: VQEL appears to be a useful optimization and initialization method for discrete emergent communication, but the current evidence does not yet justify stronger claims about language-like structure or cognitive analogies.

**Audience:**

Yes

**Audience Explanation:**

The paper should be of interest to researchers working on emergent communication, multi-agent learning, discrete latent-variable models, and representation learning. The proposed use of vector quantization as a bridge between differentiable self-play and discrete mutual communication is simple, relevant, and potentially useful.

**Broader Impact Concerns:**

I do not see major ethical concerns requiring a substantial broader impact discussion. One minor concern is that learned inter-agent communication protocols can be opaque to humans, which may matter if such methods are used in deployed multi-agent systems. A short discussion of interpretability and monitoring of learned communication protocols would be useful.

**Claims And Evidence:**

Yes

**Claims Explanation:**

The main claim that VQ-based self-play helps later communication is supported by the reported experiments. VQEL often performs better than the baselines and uses the vocabulary more fully. The ablation without self-play also supports the importance of the self-play stage.
However, I think the paper should be more careful with broader claims about language emergence or inner speech. The results show useful discrete codes for referential games, but not necessarily language-like structure in a human sense.

**Requested Changes:**

Please add more analysis of what the learned messages mean. For example, show message examples or correlations between symbols and object attributes.

Please discuss why receiver self-play performs poorly, since this limitation helps clarify when the method works.

---

> ### Author Response · Authors · 2026-06-28
>
> We sincerely thank Reviewer bh6g for the constructive feedback, thorough reading of our manuscript, and insightful suggestions. Your comments have helped us significantly improve the clarity, precision, and depth of our paper.
>
> Before addressing your specific requested changes, we would like to highlight how we addressed your broader recommendations:
>
> **A. Narrowing the cognitive claims:**
> We completely agree with your assessment that our previous framing was too strong. We have revised the **Introduction (Section 1)** to explicitly tone down the cognitive analogies.
>
> **B. Broader Impact and Interpretability:**
> We completely agree with your insightful point that the opacity of learned multi-agent protocols is a critical issue for real-world deployment. Ensuring the interpretability and monitoring of these emergent communication channels is indeed a vital consideration. We view our newly added qualitative analysis of token distributions (Appendix C) as a concrete step toward addressing this exact concern, as it provides a practical method for humans to monitor, visualize, and interpret the semantic alignments developed by the agents.
>
> Below, we address your specifically requested changes:
>
> **Response to Point 1 (Analysis of learned messages and their meaning):**
>
> We sincerely thank the reviewer for this excellent suggestion. We completely agree that understanding *what* the agents are communicating is crucial. To address this, we have added a comprehensive qualitative analysis to the revised manuscript in the new **Appendix C (Analysis of Message Compositionality)**.
>
> Specifically, we analyzed the direct correlations between generated symbols (message tokens) and underlying object attributes (shape and color) in the ShapeWorld dataset. We visualized these correlations by plotting the conditional token distributions, $P(\text{token} \mid \text{attribute value, position})$, across different methods (see **Figures 10, 11, and 12**).
>
> Our findings directly answer your question regarding what the messages mean:
> * **Baseline Methods (REINFORCE & GS-ST):** As shown in Figures 11 and 12, the token usage in the baselines is relatively diffuse. There is no strong, interpretable correlation between specific tokens and specific attributes.
> * **VQEL:** As shown in Figure 10, our method develops a highly structured and selective communication protocol. Specific symbols become strongly bound to specific semantic meanings. For example, the analysis reveals that token 4 at position 2 firmly means the color *blue*, and token 2 at position 1 is heavily correlated with the shape *rectangle*.
>
> This new analysis clearly demonstrates that VQEL not only improves raw accuracy but also induces a much more interpretable and structurally grounded language compared to the baselines. We are very grateful for your prompt, as it allowed us to showcase this important property of our method.
>
> **Response to Point 2 (Why receiver self-play performs poorly):**
>
> We thank the reviewer for this constructive suggestion — we agree that understanding why receiver self-play underperforms helps clarify the scope and mechanism of our method.
>
> The reason is rooted in how the codebook is owned and used. Successful language invention via self-play requires learning an effective codebook — the discrete vocabulary used for communication. In mutual play, this codebook is determined by the sender, since it is the sender that produces the symbolic message. When the receiver alone undergoes self-play, the codebook it develops is left unused during mutual play: the sender must still learn its own codebook from scratch. As a result, receiver self-play provides no transferable benefit, which is exactly what we observe empirically. By contrast, in sender self-play, the codebook is learned during self-play and directly reused in mutual play, making subsequent optimization substantially easier.
>
> This clarifies precisely when the method works: self-play is beneficial specifically for the agent that produces the symbolic message and thereby owns the codebook. Viewed this way, the receiver self-play result is an informative control that isolates the source of VQEL's gains — the learning of a usable codebook by the symbol-producing agent.
>
> We have expanded the discussion of this point in the main text (**Section 4.2**) and in **Appendix B** to make this explanation explicit.